# SmooSeg: Smoothness Prior for Unsupervised Semantic Segmentation

**Mengcheng Lan**[1]* **Xinjiang Wang**[3] **Yiping Ke**[2] **Jiaxing Xu**[2]
**Litong Feng**[3] **Wayne Zhang**[3]

[1] S-Lab, Nanyang Technological University
[2] SCSE, Nanyang Technological University
[3] SenseTime Research

{lanm0002, jiaxing003}@e.ntu.edu.sg    ypke@ntu.edu.sg
{wangxinjiang, fenglitong, wayne.zhang}@sensetime.com
https://github.com/mc-lan/SmooSeg

## Abstract

Unsupervised semantic segmentation is a challenging task that segments images into semantic groups without manual annotation. Prior works have primarily focused on leveraging prior knowledge of semantic consistency or priori concepts from self-supervised learning methods, which often overlook the coherence property of image segments. In this paper, we demonstrate that the smoothness prior, asserting that close features in a metric space share the same semantics, can significantly simplify segmentation by casting unsupervised semantic segmentation as an energy minimization problem. Under this paradigm, we propose a novel approach called SmooSeg that harnesses self-supervised learning methods to model the closeness relationships among observations as smoothness signals. To effectively discover coherent semantic segments, we introduce a novel smoothness loss that promotes piecewise smoothness within segments while preserving discontinuities across different segments. Additionally, to further enhance segmentation quality, we design an asymmetric teacher-student style predictor that generates smoothly updated pseudo labels, facilitating an optimal fit between observations and labeling outputs. Thanks to the rich supervision cues of the smoothness prior, our SmooSeg significantly outperforms STEGO in terms of pixel accuracy on three datasets: COCOStuff (+14.9%), Cityscapes (+13.0%), and Potsdam-3 (+5.7%).

## 1 Introduction

Semantic segmentation is a crucial task in computer vision that allows for a better understanding of the visual content and has numerous applications, including autonomous driving [1] and remote sensing imagery [2]. Despite advancements in the field, most traditional semantic segmentation models heavily rely on vast amounts of annotated data, which can be both arduous and costly to acquire. Consequently, unsupervised semantic segmentation [3; 4; 5; 6; 7; 8] has emerged as a promising alternative. Prior knowledge is fundamental to the success of unsupervised semantic segmentation models. One key prior knowledge is the principle of *semantic consistency*, which stipulates that an object's semantic label should remain consistent despite photometric or geometric transformations. Recent advances [3; 9; 4; 10] use contrastive learning to achieve consistent features or class assignments. Another essential prior knowledge is the *priori concepts* implicitly provided by self-supervised learning techniques, *e.g.*, DINO [11] and precedent arts [12; 8; 6] whose learned features can be employed to partition each image into different segments. Despite their effectiveness,

---

*Corresponding author.

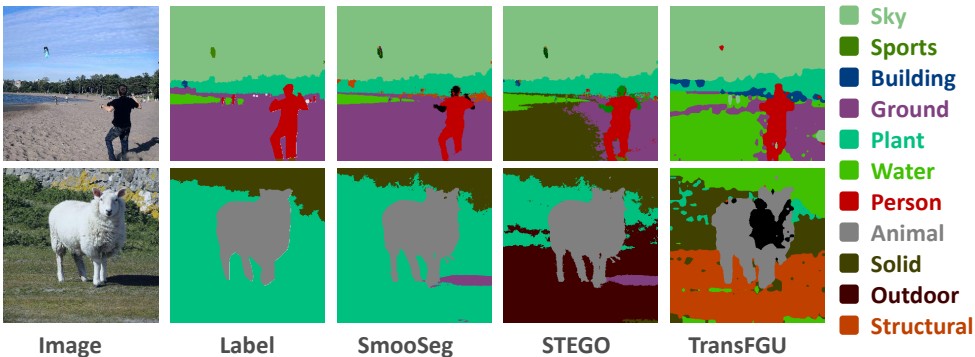

| | Sky |
| --- | --- |
| | Sports |
| | Building |
| | Ground |
| | Plant |
| | Water |
| | Person |
| | Animal |
| | Solid |
| | Outdoor |
| | Structural |

**Image**  **Label**  **SmooSeg**  **STEGO**  **TransFGU**

Figure 1: A case study of our SmooSeg with two state-of-the-arts, STEGO [8] and TransFGU [6], on the COCOStuff dataset. Our observations reveal that the segmentation maps generated by STEGO and TransFGU for regions such as the sand beach (first row) and the grassland (second row) are incomplete and lack smoothness and coherence. In contrast, our SmooSeg exhibits improved segmentation results for all these regions by considering the smoothness prior.

these methods often overlook the coherence property of image segments, resulting in predicted segments that are incomplete and lacking in coherence, as shown in Fig. 1.

Real-world images often demonstrate a natural tendency towards piecewise coherence regarding semantics, texture, or color. Observations close to each other, either in the form of adjacent pixels in the coordinate space or close features in a metric space, are expected to share similar semantic labels, and vice versa. This essential property, known as the *smoothness prior*, plays a crucial role in various computer vision tasks [13; 14; 15]. Surprisingly, it is still under-explored in the field of unsupervised semantic segmentation.

In this paper, we attempt to tackle unsupervised semantic segmentation from the perspective of *smoothness prior*. As a dense prediction task, semantic segmentation aims at finding a labeling $f \in \mathcal{F}$ that assigns each observation (pixel, patch, features) $p \in \mathcal{P}$ a semantic category $f(p)$, which could be formulated within an energy minimization framework [16]: $E(f) = E_{\text{smooth}}(f) + E_{\text{data}}(f)$. $E_{\text{smooth}}$ is a pairwise smoothness term that promotes the coherence between observations, and $E_{\text{data}}$ represents a pointwise data term that measures how well $f(p)$ fits the observation $p$. However, directly applying smoothness prior to unsupervised semantic segmentation faces several obstacles. 1) Due to the large intra-class variations in appearances within an image, it is difficult to define a well-suited similarity (dissimilarity) relationship among low-level observations. This makes it challenging to discover groups of complex observations as coherent segments. 2) $E_{\text{smooth}}$ can lead to a trivial solution where $f$ becomes smooth everywhere, a phenomenon known as model collapse. 3) Optimizing $E_{\text{data}}$ without any observed label can be challenging.

In this study, we propose a novel approach called SmooSeg for unsupervised semantic segmentation to address the aforementioned challenges. By leveraging the advantages of self-supervised representation learning in generating dense discriminate representations for images, we propose to model the closeness relationships among observations by using high-level features extracted from a frozen pre-trained model. This helps capture the underlying smoothness signals among observations. Furthermore, we implement a novel pairwise smoothness loss that encourages piecewise smoothness within segments while preserving discontinuities across image segments to effectively discover various semantic groups. Finally, we design an asymmetric teacher-student style predictor, where the teacher predictor generates smooth pseudo labels to optimize the data term, facilitating a good fit between the observations and labeling outputs.

Specifically, our model comprises a frozen feature extractor, a lightweight projector, and a predictor. The projector serves to project the high-dimensional features onto a more compact, low-dimensional embedding space, and the predictor employs two sets of learnable prototypes to generate the final segmentation results. We optimize our model using a novel energy minimization objective function. Despite its simplicity, our method has demonstrated remarkable improvements over state-of-the-art approaches. In particular, our method significantly outperforms STEGO [8] in terms of pixel accuracy on three widely used segmentation benchmarks: COCOStuff (**+14.9%**), Cityscapes (**+13.0%**), and Potsdam-3 (**+5.7%**).

## 2 Related work

**Unsupervised semantic segmentation** has gained increasing attention for automatically partitioning images into semantically meaningful regions without any annotated data. Early CRF models [17; 18] incorporate smoothness terms that maximize label agreement between similar pixels. They define adjacency for a given pixel in the coordinate space, *e.g.,* using 4-connected or 8 connected grid, which relies heavily on the low-level appearance information and falls short in capturing high-level semantic information in images. Recently, many methods [3; 4; 10] have attempted to learn semantic relationships at the pixel level with semantic consistency as a supervision signal. For example, IIC [3] is a clustering method that discovers clusters by maximizing mutual information between the class assignments of each pair of images. PiCIE [4] enforces semantic consistency between an image and its photometric and geometric augmented versions. HSG [10] achieves semantic and spatial consistency of grouping among multiple views of an image and from multiple levels of granularity. Recent advances [19; 12; 6; 8; 20] have benefited from self-supervised learning techniques, which provide priori concepts as supervision cues. For instance, InfoSeg [19] segments images by maximizing the mutual information between local pixel features and high-level class features obtained from a self-supervised learning model. The work in [12] directly employs spectral clustering on an affinity matrix constructed from the pre-trained features. TransFGU [6] generates pixel-wise pseudo labels by leveraging high-level semantic concepts discovered from DINO [11]. Additionally, STEGO [8] utilizes knowledge distillation to learn a compact representation from the features extracted from DINO based on a correspondence distillation loss, which also implies a smoothness regularization through the dimension reduction process. However, the utilization of smoothness prior in STEGO is implicit and entails separate post-process, such as min-batch K-Means, for the final semantic clustering. Besides, MaskContrast [21] and FreeSOLO [22] leverage mask priors and primarily focus on foreground object segmentation. In contrast, we propose to leverage the smoothness prior as a supervision cue to directly optimize the generated semantic map, achieving more coherent and accurate segmentation results.

**Self-supervised representation learning (SSL)** aims to learn general representations for images without additional labels, which has offered significant benefits to various downstream tasks, including detection and segmentation [6; 8]. One main paradigm of SSL is based on contrastive learning [23; 24; 25; 26; 27; 11; 28; 29], which seeks to maximize the feature similarity between an image and its augmented pairs while minimizing similarity between negative pairs. For example, MoCo [24] trains a contrastive model by using a memory bank that stores and updates negative samples in a queue-based fashion. SimCLR [23] proposes to learn a nonlinear transformation, *i.e.*, a projection head, before the contrastive loss, to improve performance. Notably, DINO [11], built upon Vision Transformer (ViT) [30], has a nice property of focusing on the semantic structure of images, such as scene layout and object boundaries. Features extracted by DINO exhibit strong semantic consistency and have demonstrated significant benefits for downstream tasks [12; 6; 8]. Another mainstream belongs to the generative learning approach [31; 32; 33]. MAE [32] and SimMIM [31] propose to predict the raw masked patches, while MaskFeat [32] proposes to predict the masked features of images. Our work also leverages recent progress in SSL for unsupervised semantic segmentation.

## 3 Method

**Problem setting.** Given a set of unannotated images $I = [I_1, \ldots, I_B] \in \mathbb{R}^{B \times 3 \times H \times W}$, where $B$ denotes the number of images, and $3, H, W$ represent the channel, height, and width dimensions respectively, the objective of unsupervised semantic segmentation is to learn a labeling function $f \in \mathcal{F}$ that predicts the semantic label for each pixel in each image. We represent the predicted semantic maps as $Y = [Y_1, \ldots, Y_B] \in \{1, \cdots, K\}^{B \times H \times W}$, where $K$ refers to the number of predefined categories.

**Architecture.** To achieve this goal, we introduce the SmooSeg approach, which capitalizes on self-supervised representation learning and smoothness prior within an energy minimization framework, as illustrated in Fig. 2. SmooSeg comprises three primary components: a feature extractor $f_\theta$, a projector $h_\theta$, and a predictor $g_\theta$. Initially, for each image $I_i$, we employ a pre-trained backbone network, such as a frozen version of DINO, to acquire feature representations $X_i = f_\theta(I_i) \in \mathbb{R}^{C \times N}$, where $C$ and $N$ denote the number of feature channels and image patches, respectively. Subsequently, the projector $h_\theta$ maps these features onto a low-dimensional embedding space, resulting in a set

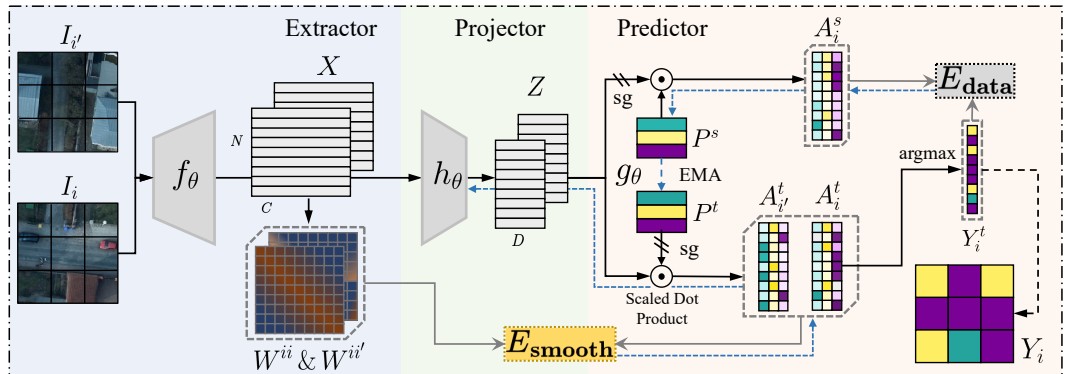

Figure 2: Overview of our SmooSeg framework, showing the application of the smoothness prior within image $I_i$ and across images $I_i$ and $I_{i'}$. sg denotes the stop-gradient operation.

of compact features $Z_i = h_\theta(X_i) \in \mathbb{R}^{D \times N}$, where $D$ denotes the reduced feature dimensionality. Finally, the predictor $g_\theta$ generates the label assignments $A_i^{\{s,t\}} \in \mathbb{R}^{K \times N}$ by computing the similarity scores between the compact features $Z_i$ and the prototypes $P^{\{s,t\}}$. Here, $P^s$ and $P^t$ represent student and teacher prototypes, respectively. The semantic map $Y_i$ for image $I_i$ can be obtained by reshaping the output $Y_i^t$ of the teacher branch.

## 3.1 Smoothness Prior

Real-world images typically exhibit inherent continuity and coherence in terms of semantics, texture, and color. Within a single object, semantic labels tend to demonstrate smoothness and consistency, ensuring a cohesive representation of the object. In contrast, labels between distinct objects manifest discontinuity and divergence, facilitating the separation of different object instances. This essential property, known as the smoothness prior, is expected to play a critical role in guiding unsupervised semantic segmentation tasks toward more accurate and meaningful segmentation results. We therefore consider the following pairwise smoothness term:

$$E_{\text{smooth}} = \sum_{i=1}^{B} \sum_{p,q=1}^{N} W_{pq}^{ii} \cdot \delta(Y_{i,p}, Y_{i,q}), \tag{1}$$

where $W^{ii} \in \mathbb{R}^{N \times N}$ is the closeness matrix of image $I_i$. $\delta(Y_{i,p}, Y_{i,q})$ is the penalty that takes the value of 1 if $Y_{i,p} \neq Y_{i,q}$, and 0 otherwise. By minimizing this smoothness term, two close patches with different labels will be penalized. In other words, the segmentation model is encouraged to assign similar labels to close patches, thereby promoting the coherence within objects.

**Closeness matrix.** It is worth noting that the large intra-class variation in appearances within the raw pixel space renders the discovery of well-suited closeness relationships among low-level observations challenging. We therefore propose to model the closeness relationships by the cosine distance in the high-level feature space. Specifically, $W^{ii}$ can be calculated by:

$$W_{pq}^{ii} = \frac{X_{i,p} \cdot X_{i,q}}{\|X_{i,p}\| \|X_{i,q}\|}, \tag{2}$$

where $X_{i,p}$ and $X_{i,q}$ represent the feature vectors for patches $p$ and $q$ of image $I_i$, respectively. Theoretically, a large element value in the closeness matrix, *i.e.,* a high cosine similarity, suggests a high possibility of a close patch pair, and vice versa. We apply a zero-mean normalization to this matrix: $\bar{W}_p^{ii} = W_p^{ii} - \frac{1}{N}\sum_q W_{pq}^{ii}$. This normalization balances the negative and positive forces during optimization, which prevents excessive influence from either the negative or positive components of the closeness matrix and ensures that the optimization process is more stable.

**Label penalty.** Directly minimizing Eq. 1 to optimize our segmentation model is not feasible due to the non-differentiable property of $\delta(\cdot, \cdot)$ and the hard label assignment $Y$. As a result, we have to

resort to another form of penalty cost. Suppose we have the soft label assignment $A_i^t \in \mathbb{R}^{K \times N}$ of image $I_i$ (which will be introduced later), by which we can redefine the penalty cost function as:

$$\delta(A_{i,p}^t, A_{i,q}^t) = 1 - \frac{A_{i,p}^t \cdot A_{i,q}^t}{\|A_{i,p}^t\| \|A_{i,q}^t\|}. \tag{3}$$

Because the non-negative property of the softmax output, *i.e.*, $0 \leq A^t$, $0 \leq \delta(\cdot, \cdot) \leq 1$ always holds. A larger value of $\delta(\cdot, \cdot)$ denotes a greater dissimilarity between two labels, thereby indicating a higher penalty cost, and vice versa.

**Smoothness prior within and across images.** To prevent the model from converging to a trivial solution where the labeling function becomes smooth everywhere, we also apply the smoothness prior across images, acting as a strong negative force, by introducing another image $I_{i'}$ that is randomly selected from the current batch. We then obtain the final smoothness term:

$$E_{\text{smooth}} = E_{\text{smooth}}^{\text{within}} + E_{\text{smooth}}^{\text{across}} = \sum_{i=1}^{B} \sum_{p,q=1}^{N} \{(\bar{W}_{pq}^{ii} - b_1) \cdot \delta(A_{i,p}^t, A_{i,q}^t) + (\bar{W}_{pq}^{ii'} - b_2) \cdot \delta(A_{i,p}^t, A_{i',q}^t)\}. \tag{4}$$

Here, we introduce a scalar $b_1$ to adjust the threshold for applying the penalty. That is, when $\bar{W}_{pq}^{ii} - b_1 > 0$, indicating that two patches $p, q$ with a high closeness degree are nearby patches in the embedding space, patches $p, q$ with different labels will be penalized, encouraging the piecewise smoothness within segments; otherwise, they are rewarded to assign different labels, leading to the discontinuities across segments. By doing so, SmooSeg is capable of finding globally coherent semantic segmentation maps.

**Discussion with CRF and STEGO.** CRF methods [17; 18] model the closeness relationship of pixels using their spatial coordinates, emphasizing the local smoothness within each image. On the contrary, our SmooSeg encodes the global closeness relationship of image patches based on the cosine distance in the feature space, which can discover the high-level semantic groups of images. Our smoothness term appears to be similar to the correlation loss in STEGO: $\mathcal{L}_{\text{corr}} = -\sum(F - b) \max(S, 0)$, but essentially the two losses model different things. In STEGO, $S$ denotes the feature correlation, by which STEGO aims to learn low-dimensional compact representations for images through a learnable projection head. A separate clustering algorithm, *e.g.*, k-means, is required to obtain the final segmentation maps. However, even with the learned compact representations, the coherence of image segments is not guaranteed in STEGO as slight differences in features may lead to inconsistent labels in the clustering stage. In contrast, our SmooSeg aims to directly learn a labeling function (projector + predictor) based on the smoothness prior, which encourages piecewise smoothness within segments and preserves disparities across segments, leading to more coherent and semantically meaningful segmentation maps. Additionally, the negative part of $S$ contradicts the learning intention of STEGO and therefore requires a 0-clamp via $\max(S, 0)$, which however, represents discontinuities between image patches and should be preserved. In contrast, our label penalty $0 \leq \delta(\cdot, \cdot) \leq 1$ has a desirable property compared to $S$.

## 3.2 Asymmetric Predictor

A desirable labeling function learnt through energy minimization should on the one hand produce piecewise smooth results, and on the other hand be well fit between the observations and labeling outputs. For semantic segmentation, we expect the labeling output of an image to align well with its semantic map. In other words, the labeling output should accurately predict a category for each individual pixel with high confidence or low entropy. However, this goal is a nutshell in unsupervised semantic segmentation as there is no observed semantic map.

Self-training [34; 29] emerges as a promising solution for tasks involving unlabeled data. To address the above challenge, we design an asymmetric student-teacher style predictor to learn the labeling function through a stable self-training strategy. The student branch employs a set of $K$ learnable prototypes (class centers) $P^s = [p_1^s, \cdots, p_K^s] \in \mathbb{R}^{K \times D}$ to predict the semantic maps of images. The teacher branch holds the same number of prototypes $P^t$ as the student, and $P^t$ is updated as an exponential moving average of $P^s$. We then compute the soft assignment $A_i^{\{s,t\}}$ of the embeddings $Z_i$ with the prototypes $P^{\{s,t\}}$ by computing their cosine similarity. With $\ell_2$-normalized embeddings

**Algorithm 1** SmooSeg: PyTorch-like Pseudocode

```
# f, h: extractor, projector
# Ps, Pt, tau, alpha: student prototypes, teacher prototypes, temperature, momentum

for img in dataloader:
    x = f(img)                                          # feature extraction
    z = h(x)                                            # feature projection
    {x, z, Ps, Pt} = F.normalize({x, z, Ps, Pt}, dim=1)
    As = matmul(z.T.detach(), Ps)                    # student label assignment
    At = softmax(matmul(z.T, Pt.detach())/tau, dim=1) # teacher label assignment
    loss_data = F.cross_entropy(As, argmax(At))

    At = F.normalize(At, dim=1)
    {x_p, At_p} = shuffle({x, At})                # shuffle along the batch dimension
    W = matmul(x.T, x) - matmul(x.T, x).mean(dim=-1)
    W_p = matmul(x.T, x_p) - matmul(x.T, x_p).mean(dim=-1)
    delta = 1 - matmul(At.T, At)
    delta_p = 1 - matmul(At.T, At_p)
    loss_smooth = sum((W - b1) * delta + (W_p - b2) * delta_p)

    (loss_smooth + loss_data).backward()
    update(h, Ps)                           # update projector and student prototypes
    Pt = alpha * Pt + (1 - alpha) * Ps          # momentum update teacher prototypes
```

$\bar{Z}_i = Z_i/\|Z_i\|$ and prototypes $\bar{P}^{\{s,t\}} = P^{\{s,t\}}/\|P^{\{s,t\}}\|$, we have

$$A_i^s = \text{softmax}(\bar{P^s} \cdot \text{sg}(\bar{Z}_i)), \quad A_i^t = \text{softmax}((\text{sg}(\bar{P^t}) \cdot \bar{Z}_i)/\tau) \in \mathbb{R}^{K \times N}, \tag{5}$$

where temperature parameter $\tau > 0$ controls the sharpness of the output distribution of the teacher branch. The teacher branch is responsible for generating smoothly updated pseudo labels to supervise the student prototypes' learning. By using a patch-wise cross-entropy loss, we have the data term as

$$E_{\text{data}} = -\sum_{i=1}^{B} \sum_{p=1}^{N} \sum_{k=1}^{K} \mathbb{I}_{Y_{i,p}^t = k} \log A_{i,p,k}^s, \tag{6}$$

where $\mathbb{I}_\cdot$ is an indicator that outputs 1 if the argument is true, and 0 otherwise. $Y_i^t = \text{argmax } A_i^t$ is the hard pseudo label for patch $p$ of image $I_i$. By minimizing $E_{\text{data}}$, the segmentation model is expected to generate label assignments for each patch with high confidence, thus ensuring a better fit between the observations and their predicted labels.

### 3.3 Overall Optimization Objective

Our final optimization objective function for training SmooSeg is obtained by incorporating the smoothness term and the data term as follows:

$$\mathcal{L} = \sum_{i=1}^{B} \sum_{p,q=1}^{N} \{(\bar{W}_{pq}^{ii} - b_1) \cdot \delta(A_{i,p}^t, A_{i,q}^t) + (\bar{W}_{pq}^{ii'} - b_2) \cdot \delta(A_{i,p}^t, A_{i',q}^t)\}$$
$$- \sum_{i=1}^{B} \sum_{p=1}^{N} \sum_{k=1}^{K} \mathbb{I}_{Y_{i,p}^t = k} \log A_{i,p,k}^s. \tag{7}$$

In practice, $\mathcal{L}$ could be approximately minimized using Stochastic Gradient Descent (SGD). During each training iteration, the projector is optimized using gradients from the smoothness loss, while the student prototypes are optimized using gradients from the data loss. The teacher prototypes are updated as an exponential moving average of the student prototypes: $P^t = \alpha P^t + (1 - \alpha)P^s$, with $\alpha$ denoting the momentum value. After training, we use the output from the teacher branch as the segmentation results. The overall procedure in pytorch-like pseudocode of SmooSeg is summarized in Algorithm 1.

# 4 Experiments

## 4.1 Experimental Setup

**Datasets.** Our experimental setup mainly follows that in previous works [8; 4] in datasets and evaluation protocols. We test on three datasets. **COCOStuff** [35] is a scene-centric dataset with a total of 80 things and 91 stuff categories. Classes are merged into 27 categories for evaluation, including 15 stuff and 12 things. **Cityscapes** [36] is a collection of street scene images from 50 cities, with classes merged into 27 classes by excluding the "void" class. **Potsdam-3** [3] is a remote sensing dataset with 8550 images belonging to 3 classes, in which 4545 images are used for training and 855 for testing.

**Evaluation metrics.** For all models, we utilize the Hungarian matching algorithm to align the prediction and the ground-truth semantic map for all images. We also use a CRF [17; 8] as the post-processing to refine the predicted semantic maps. Two quality metrics including mean Intersection over Union (**mIoU**) and Accuracy (**Acc**) over all the semantic categories are used in the evaluation.

**Implementation details.** Our experiments were conducted using PyTorch [37] on an RTX 3090 GPU. To ensure a fair comparison with previous works [6; 8], we use DINO [11] with a ViT-small $8 \times 8$ backbone pre-trained on ImageNet as our default feature extractor, which is frozen during model training. Our projector consists of a linear layer and a two-layer SiLU MLP whose outputs are summed together. The predictor contains two sets of prototypes with the same initialization. The exponential moving average (EMA) hyper-parameter is set to $\alpha = 0.998$. The dimension of the embedding space is $D = 64$. The temperature is set to $\tau = 0.1$. We use the Adam optimizer [38] with a learning rate of $1 \times 10^{-4}$ and $5 \times 10^{-4}$ for the projector and predictor, respectively.

We set a batch size of 32 for all datasets. For Cityscapes and COCOStuff datasets, we employ a five-crop technique to augment the training set size. We train our model with a total of 3000 iterations for Cityscapes and Potsdam-3 datasets, and 8000 iterations for the COCOStuff dataset.

## 4.2 Comparison with State-of-the-Arts

**Quantitative results.** We summarise the quantitative results on three datasets in Tables 1, 2 and 3, respectively. Results of baselines, ResNet50[39], MoCoV2[40] and DINO[11] are directly cited from the paper [8], while the results of DINOV2 [41] (Table 3) are obtained by our implementation. For these baselines, we first extracted dense features for all images, then utilized a minibatch k-means algorithm to perform patches grouping, which resulted in the final segmentation maps. Our SmooSeg significantly outperforms all the state-of-the-art methods in terms of both pixel accuracy and mIoU on all datasets. In particular, on the CO-COStuff dataset in Table 1, with DINO ViT-S/8 as backbone, SmooSeg gains a 14.9% improvement in pixel accuracy and a 2.2% improvement in mIoU over the best-performing baseline STEGO.

We observe that TransFGU outperforms STEGO in terms of accuracy, but is inferior in mIoU on both COCOStuff and Cityscapes. This is due to

Table 1: Performance on the COCOStuff dataset (27 classes).

| Methods | backbone | Acc. | mIoU |
|---|---|---|---|
| ResNet50 [39] | ResNet50 | 24.6 | 8.9 |
| IIC [3] | R18+FPN | 21.8 | 6.7 |
| MDC [4] | R18+FPN | 32.2 | 9.8 |
| PiCIE [4] | R18+FPN | 48.1 | 13.8 |
| PiCIE+H [4] | R18+FPN | 50.0 | 14.4 |
| SlotCon [29] | ResNet50 | 42.4 | 18.3 |
| MoCoV2 [40] | ResNet50 | 25.2 | 10.4 |
| + STEGO [8] | ResNet50 | 43.1 | 19.6 |
| + SmooSeg | ResNet50 | 52.4 | 18.8 |
| DINO [11] | ViT-S/8 | 29.6 | 10.8 |
| + TransFGU [6] | ViT-S/8 | 52.7 | 17.5 |
| + STEGO [8] | ViT-S/8 | 48.3 | 24.5 |
| + SmooSeg | ViT-S/8 | **63.2** | **26.7** |

Table 2: Performance on the Cityscapes Dataset (27 Classes).

| Methods | backbone | Acc. | mIoU |
|---|---|---|---|
| IIC [3] | R18+FPN | 47.9 | 6.4 |
| MDC [4] | R18+FPN | 40.7 | 7.1 |
| PiCIE [4] | R18+FPN | 65.5 | 12.3 |
| DINO [11] | ViT-S/8 | 40.5 | 13.7 |
| + TransFGU [6] | ViT-S/8 | 77.9 | 16.8 |
| + STEGO [8] | ViT-S/8 | 69.8 | 17.6 |
| + SmooSeg | ViT-S/8 | **82.8** | **18.4** |

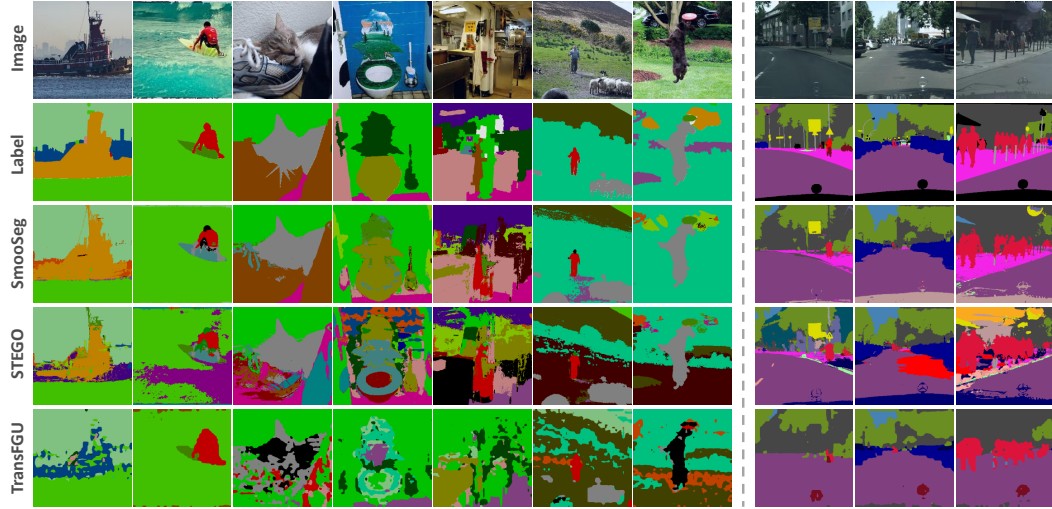

Figure 3: Qualitative results on COCOStuff (left) and Cityscapes (right) datasets.

Table 3: Performance on the Potsdam-3 Dataset.

| Methods | backbone | Acc. | mIoU |
|---|---|---|---|
| Random CNN [3] | VGG11 | 38.2 | - |
| K-means [42] | VGG11 | 45.7 | - |
| SIFT [43] | VGG11 | 38.2 | - |
| IIC [3] | R18+FPN | 65.1 | - |
| Deep Cluster [44] | R18+FPN | 41.7 | - |
| InfoSeg [19] | CNN | 71.6 | - |
| DINO [11] | ViT-B/8 | 62.2 | 43.3 |
| + STEGO [8] | ViT-B/8 | 77.0 | 62.6 |
| + SmooSeg | ViT-B/8 | **82.7** | **70.3** |
| DINOV2 [41] | ViT-B/14 | 81.8 | 69.0 |
| + SmooSeg | ViT-B/14 | 86.3 | 75.7 |

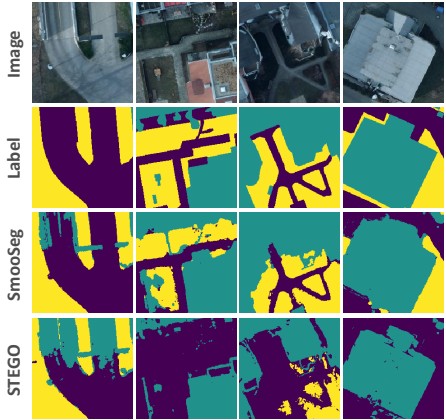

Figure 4: Qualitative results on Postdam-3 dataset.

the fact that TransFGU adopts a pixel-wise cross-entropy loss, which focuses more on the overall accuracy of pixels, while STEGO achieves better class-balanced segmentation results through mini-batch k-means. Our SmooSeg significantly outperforms both TransFGU and STEGO in both accuracy and mIoU, We attribute this superiority to our energy minimization loss, which optimizes both the smoothness term and the data term simultaneously.

Same conclusions can be drawn on the Postdam-3 dataset, as shown in Table 3. We can see that SmooSeg, with DINO ViT-B/8 as the backbone, significantly outperforms STEGO, with gains of 5.7% in accuracy and 7.7% in mIoU. The improvement is particularly significant in terms of mIoU. This is not surprising as Potsdam-3 is a remote sensing image dataset that contains only 3 classes, so segments on the Potsdam-3 are often relatively large. In such a scenario, the smoothness prior becomes even more important in ensuring coherent segmentation maps.

**Qualitative results.** We present qualitative examples of SmooSeg, STEGO and TransFGU on three datasets in Figs. 3 and 4. Additional qualitative results, along with color maps, can be found in Appendix C. As shown in Fig. 3, SmooSeg produces high-quality fine-grained segmentation maps that outperform those obtained by STEGO and TransFGU. Though STEGO uses a feature correspondence loss to encourage features to form compact clusters, its segmentation maps still suffer from incoherence problems. It can be observed that, slight differences in features can result in inconsistent labels in STEGO. The second image from the left column of Fig. 3 shows such an example: the slight variations in light on the sea surface results in differences in features, leading to an

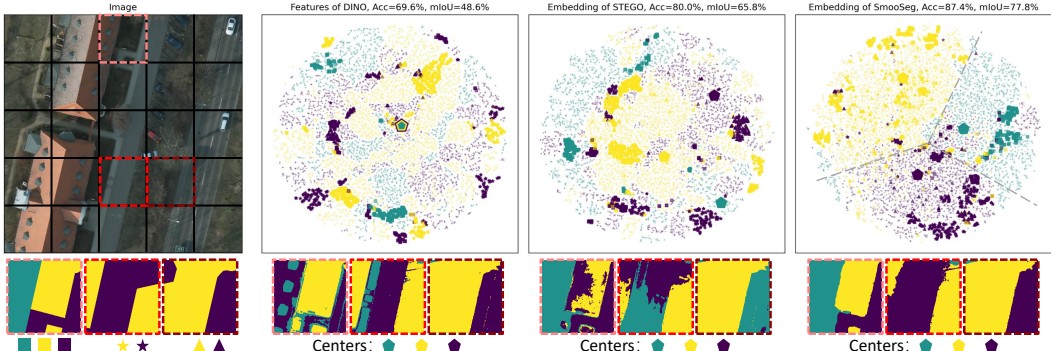

Figure 5: t-SNE [45] visualization of feature embeddings for DINO, STEGO, and our SmooSeg on the Potsdam-3 dataset. The large-scale image is partitioned into 25 sub-images with a resolution of $224 \times 224$ before being input into each algorithm. We highlight the distribution of three sub-images using square, star and triangle_up markers, respectively, and plot the class centers of each algorithm using three pentagon markers with different colors, where teal, yellow and purple represent *buildings + clutter*, *roads + cars*, *vegetation + trees*, respectively. Best viewed in color and zoomed in.

incomplete and incoherent water segment. Similar phenomenon can be observed in the segmentation maps on Cityscapes and Potsdam-3 too. Besides, although TransFGU is a top-down approach, it still overlooks the relationship between image patches in its top-down approach, and therefore achieves much worse segmentation results. In contrast, SmooSeg with the aim of generating smooth label assignments within segments while preserving differences across different segments by leveraging the smoothness prior, the semantic maps produced by SmooSeg show more coherent and semantically meaningful results. In Fig. 4, we can see that SmooSeg outperforms the other methods in terms of accurate boundaries.

### 4.3 Analyses

**Visualization.** Feature visualizations of DINO, STEGO and SmooSeg are illustrated in Fig. 5. We can see that the feature distribution of DINO with ViT-base/8 as the backbone exhibits some semantic consistency, with compact clusters within each image but disperse across images. The embeddings of STEGO, which are distilled from DINO features using feature correspondence loss, show higher semantic consistency than DINO, with more compact clusters across images, such as the yellow markers, and improved performance. However, STEGO still suffers from the label incoherence problem due to the large intra-class variation of embeddings, indicating that feature distillation alone is insufficient to capture the high-level semantic coherence of segments. Our SmooSeg leverages the smoothness prior to encourage smooth label assignments, measured by the cosine distance between patch embeddings and prototypes (centers), and achieves remarkable improvement in the semantic consistency of feature embeddings. As shown in the right part of Fig. 5, SmooSeg produces highly semantically compact and coherent clusters with clear class boundaries for all images, and the performance, at 87.4% Acc and 77.8% mIoU, significantly higher than STEGO. These results further prove the effectiveness of our SmooSeg in using smoothness prior for unsupervised semantic segmentation.

**Objective function.** To assess the effectiveness of our energy minimization objective function, we conduct an ablation study on the COCOStuff dataset by comparing SmooSeg with four variants of the objective function, each with a different term removed. For the variant of w/o $E_{\text{data}}$, we only keep the prototypes in the teacher branch to generate the label map for the smoothness term. The results are shown in Table 4, where $E_{\text{smooth}}^{\text{across}}$ denotes the smoothness term across different images. We can see that the performance drops by 10.2% on Acc and 1.2% on mIoU when removing the data term $E_{\text{data}}$, which highlights the importance of the data term in promoting the fitting of the labeling function. Besides, removing $E_{\text{smooth}}^{\text{across}}$ results in a much larger drop in performance, with a decrease of 27.1% on Acc and 16.3% on mIoU. Moreover, our segmentation model fails when removing the entire $E_{\text{smooth}}$. The smoothness term utilizes a closeness matrix constructed from high-level features of a pre-trained model, acting as strong supervision signals to guide the label learning for all

Table 4: Analysis of objective function on the COCOStuff dataset.

| Methods | Acc. | mIoU |
|---|---|---|
| SmooSeg | 63.2 | 26.7 |
| w/o $E_{\text{data}}$ | 53.0 | 25.5 |
| w/o $E_{\text{smooth}}^{\text{across}}$ | 36.1 | 10.4 |
| w/o $E_{\text{smooth}}$ | 16.8 | 0.6 |

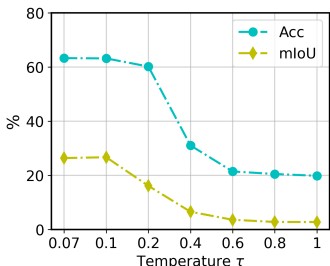

Figure 6: Sensitivity study on the temperature parameter $\tau$.

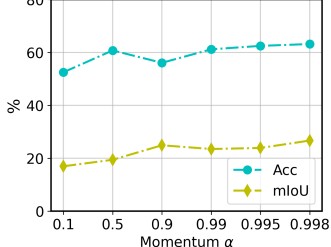

Figure 7: Sensitivity study on the momentum parameter $\alpha$.

image patches. Therefore, it is reasonable to see that $E_{\text{smooth}}$ contributes significantly to the overall performance. On the contrary, the data term operates in a self-training fashion with pseudo labels derived from the teacher branch, which alone cannot generate accurate segmentation maps. These findings demonstrate the crucial role of both the data and smoothness terms for optimal performance of SmooSeg in unsupervised semantic segmentation.

**Temperature parameter $\tau$.** We investigate the effect of the temperature parameter $\tau$ on the performance of SmooSeg on the COCOStuff dataset, and report the results in Fig. 6. Theoretically, a smaller $\tau$ sharpens the softmax output, providing greater gradients and supervision signals for model training. Fig. 6 shows that $\tau$ plays a critical factor in the success of SmooSeg. Specifically, SmooSeg achieves good results when $\tau \leq 0.1$, while performance drops considerably when $\tau \geq 0.2$ because the softmax output tends to become uniformly distributed.

**Momentum parameter $\alpha$.** We also study the impact of the $\alpha$ on SmooSeg. $\alpha$ controls the smoothness of the update of the teacher predictor from the student predictor. We plot the performance on the COCOStuff dataset as $\alpha$ changes from 0.1 to 1 in Fig. 7. The performance of SmooSeg gradually improves as $\alpha$ increases, and reaches stable when $0.99 \leq \alpha$.

**Limitation.** Setting hyper-parameters without cross-validation is always a challenge for unsupervised learning methods. The main limitation of our method is that it involves two dataset-specific hyper-parameters in the smoothness term. We present a feasible strategy in Appendix A to alleviate this issue.

## 5 Conclusions

In this paper, we propose SmooSeg, a simple yet effective unsupervised semantic segmentation approach that delves into the potential of the smoothness prior, emphasizing the coherence property of image segments. In particular, we implement a pairwise smoothness loss to effectively discover semantically meaningful groups. We also design an asymmetric teacher-student style predictor to generate high-quality segmentation maps. SmooSeg comprises a frozen extractor, as well as a lightweight projector and a predictor which could be optimized using our energy minimization objective function. Experimental results show that SmooSeg outperforms state-of-the-art approaches on three widely used segmentation benchmarks by large margins.

**Acknowledgement.** This research is supported under the RIE2020 Industry Alignment Fund – Industry Collaboration Projects (IAF-ICP) Funding Initiative, as well as cash and in-kind contribution from the industry partner(s), by the National Research Foundation, Singapore under its Industry Alignment Fund – Pre-positioning (IAF-PP) Funding Initiative, and by the Ministry of Education, Singapore under its MOE Academic Research Fund Tier 2 (STEM RIE2025 Award MOE-T2EP20220-0006). Any opinions, findings and conclusions or recommendations expressed in this material are those of the author(s) and do not reflect the views of National Research Foundation, Singapore, and the Ministry of Education, Singapore.

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

# Appendix

## A  Setting Hyper-parameters

Tuning hyperparameters without cross-validation on labels is particularly challenging for unsupervised learning methods. For example, STEGO [8] contains six parameters that are both dataset- and network-specific. Inspired by their hyperparameter tuning methods, we introduce a similar strategy for manual hyperparameter tuning. Recall that our SmooSeg introduces two hyperparameters $b_1$ and $b_2$ in the smoothness loss, which are used to adjust the threshold for applying the penalty. Ideally, a segmentation model should promote smoothness within segments while maintaining discontinuity between different segments. To achieve this, we can monitor the distribution of $\delta$, referred to as the smoothness degree, during the model training (see Fig. 8). In a balanced setting, the smoothness degree distribution should exhibit bimodality, as demonstrated in the first column of Fig. 8. Otherwise, the segmentation model will generate semantic maps that are either too smooth or too discontinuous. Specifically, The hyperparameters used in SmooSeg with DINO as the backbone are summaried in Table 5.

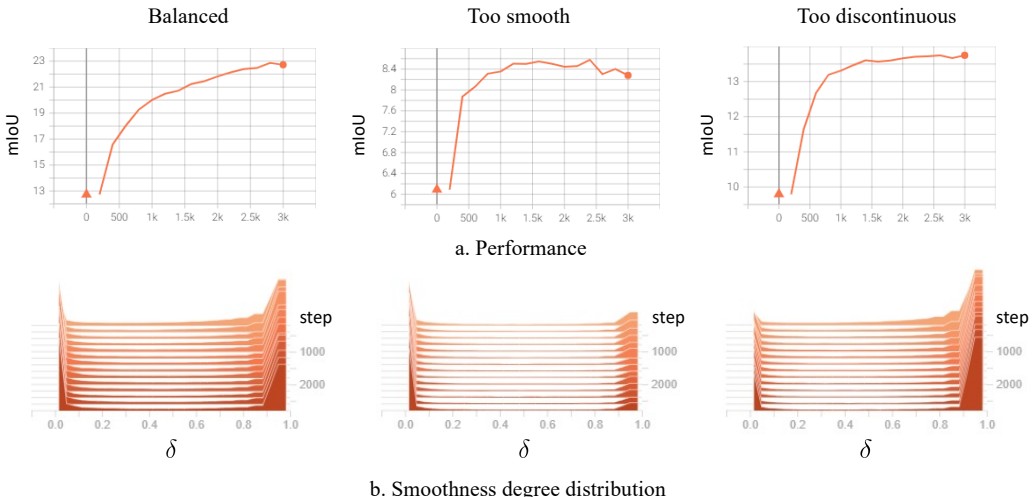

Figure 8: Distributions of the label penalty $\delta = 1 - \cos(A, A)$ within an image and their corresponding performance under three different parameter settings. If the distribution of $\delta$ tends toward single-modal with peaks at $0.0$ (or $1.0$), the semantic maps will be overly smooth (or discontinuous).

Table 5: Hyperparameters used in SmooSeg.

| Parameter | COCOStuff | Cityscapes | Potsdam-3 |
|---|---|---|---|
| $b_1$ | 0.5 | 0.5 | 0.5 |
| $b_2$ | -0.02 | -0.02 | 0.1 |

## B  The impact of CRF

CRF postprocessing is a common practice in both supervised and unsupervised semantic segmentation [8], and that the use of CRF does not overshadow the contribution of the smoothness term in our work. The smoothness prior in this work performs on the high-level feature maps and mainly contributes to semantic smoothness, while CRF operates on pixels to refine the fine details and remedy the resolution loss caused by the final upsample operation that exists in most semantic segmentation models (a normal upsample rate is 8x8). Therefore, the application of a CRF serves as a supplement to our smoothness prior to further refine low-level smoothness.

Table 6: Experimental results of the impact of CRF on SmooSeg and STEGO.

| | COCOStuff | | Cityscapes | | Potsdam-3 | | Avg. | |
|---|---|---|---|---|---|---|---|---|
| | Acc | mIoU | Acc | mIoU | Acc | mIoU | Acc | mIoU |
| STEGO w/o CRF | 46.5 | 22.4 | 63.5 | 16.8 | 74.1 | 58.9 | 61.4 | 32.7 |
| STEGO w CRF | 48.3 | 24.5 | 69.8 | 17.6 | 77.0 | 62.6 | 65.0 (+3.6) | 34.9 (+2.2) |
| SmooSeg w/o CRF | 60.6 | 25.2 | 79.8 | 18.0 | 81.4 | 68.4 | 73.9 | 37.2 |
| SmooSeg w CRF | 63.2 | 26.7 | 82.8 | 18.4 | 82.7 | 70.3 | 76.2 (+2.3) | 38.5 (+1.3) |

Table 6 demonstrates that SmooSeg still achieves state-of-the-art without CRF. Overall, the performance degradation in STEGO is notably more pronounced compared to SmooSeg. Importantly, the application of a CRF serves as an effective supplement to our smoothness prior. Moreover, we also present qualitative visualizations with and without CRF in Figures 9 and 10. It is found that CRF is able to refine the quality of fine details on both STEGO and SmooSeg. However, SmooSeg is consistently more semantically coherent than STEGO either with or without CRF.

## C  Additional Qualitative Results

To provide further evaluation, we have included some difficult samples from the COCOStuff dataset that were predicted by our Smoseg and STEGO in Figure 11. In these cases, Smoseg and STEGO tend to generate inaccurate semantic maps. However, it is noteworthy that even in challenging scenarios, SmooSeg consistently generates more semantically coherent segmentation maps compared to STEGO. This observation underscores the advantages of incorporating our smoothness prior in semantic segmentation tasks.

We provide the visualization of more results in Fig. 12 for the COCOStuff dataset, Fig 13 for the Cityscapes dataset, Figs 14 and 15 for the Potsdam-3 datasets. Overall, SmooSeg consistently produces more coherent segmentation maps when compared to both STEGO and TransFGU. These visual results provide further evidence of the efficacy of the smoothness prior in enhancing the label coherence and the overall segmentation quality in unsupervised settings.

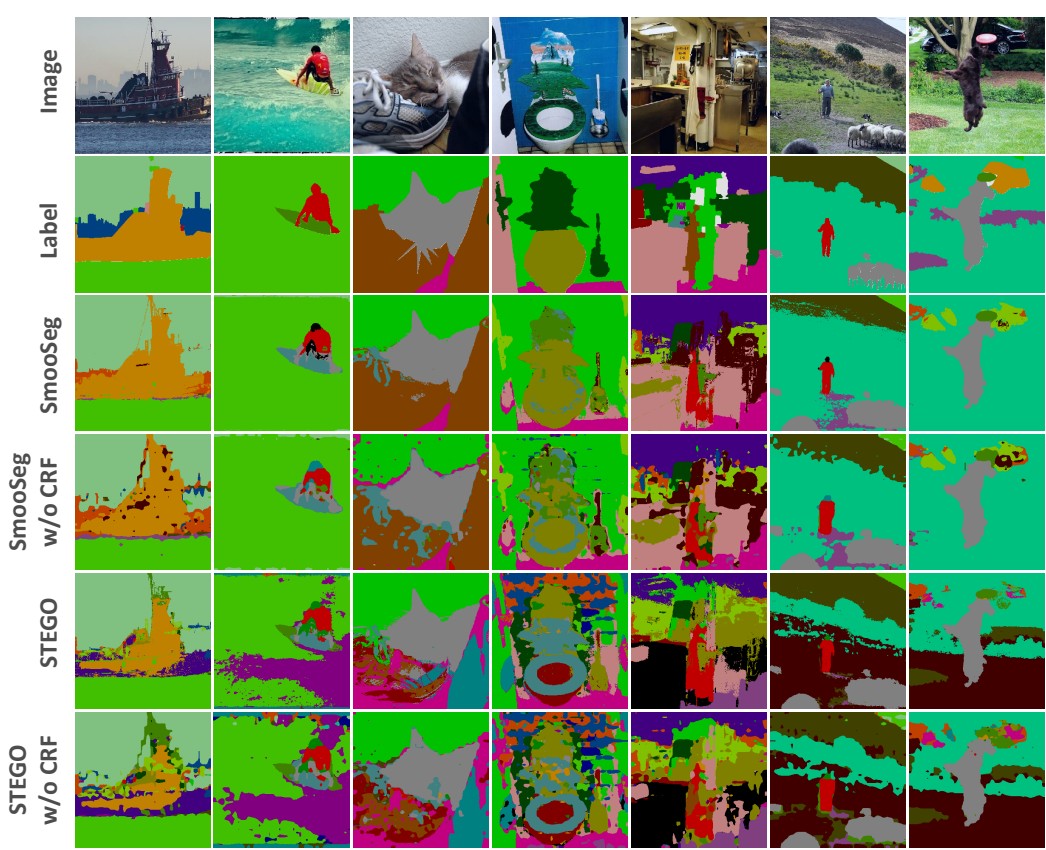

Figure 9: Qualitative results of STEGO and SmooSeg with and without CRF on the COCOStuff dataset.

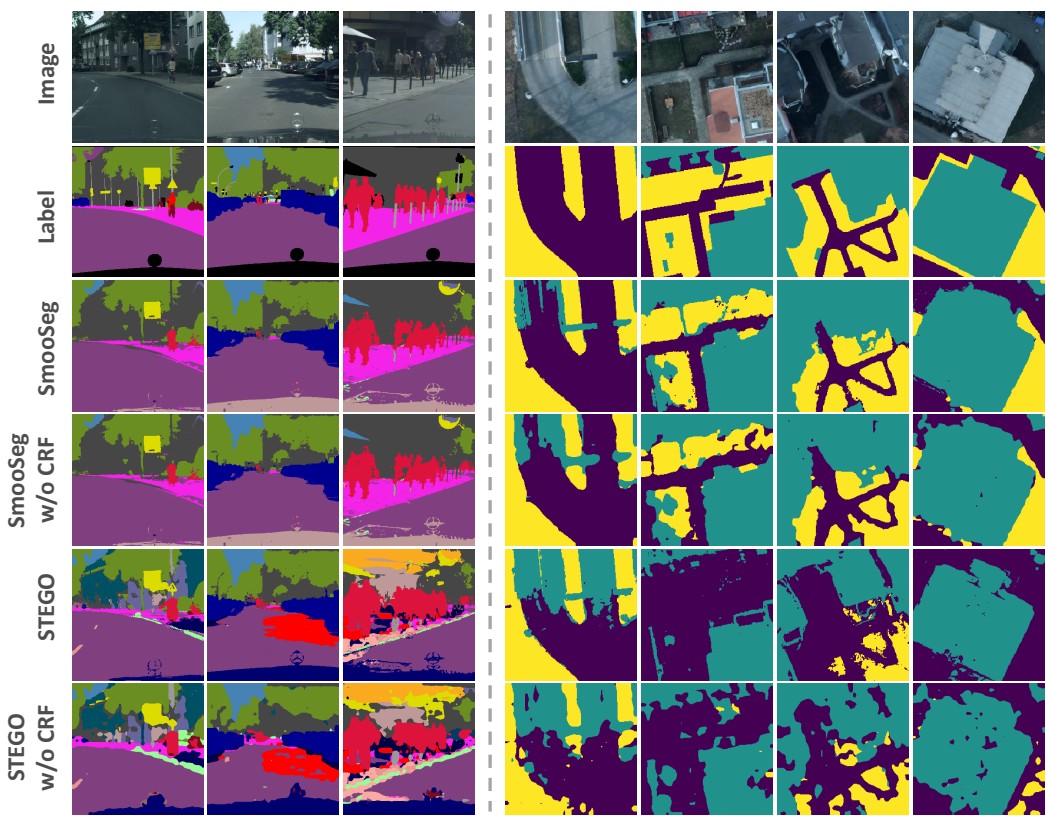

Figure 10: Qualitative results of STEGO and SmooSeg with and without CRF on the Cityscapes and Potsdam-3 datasets.

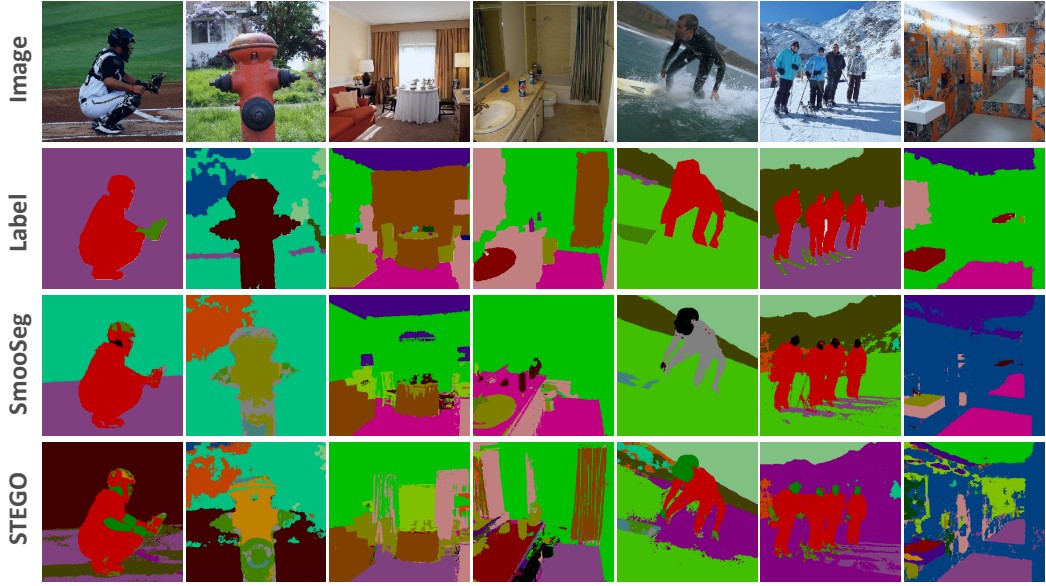

Figure 11: Difficult examples predicted by SmooSeg and STEGO on the COCOStuff dataset.

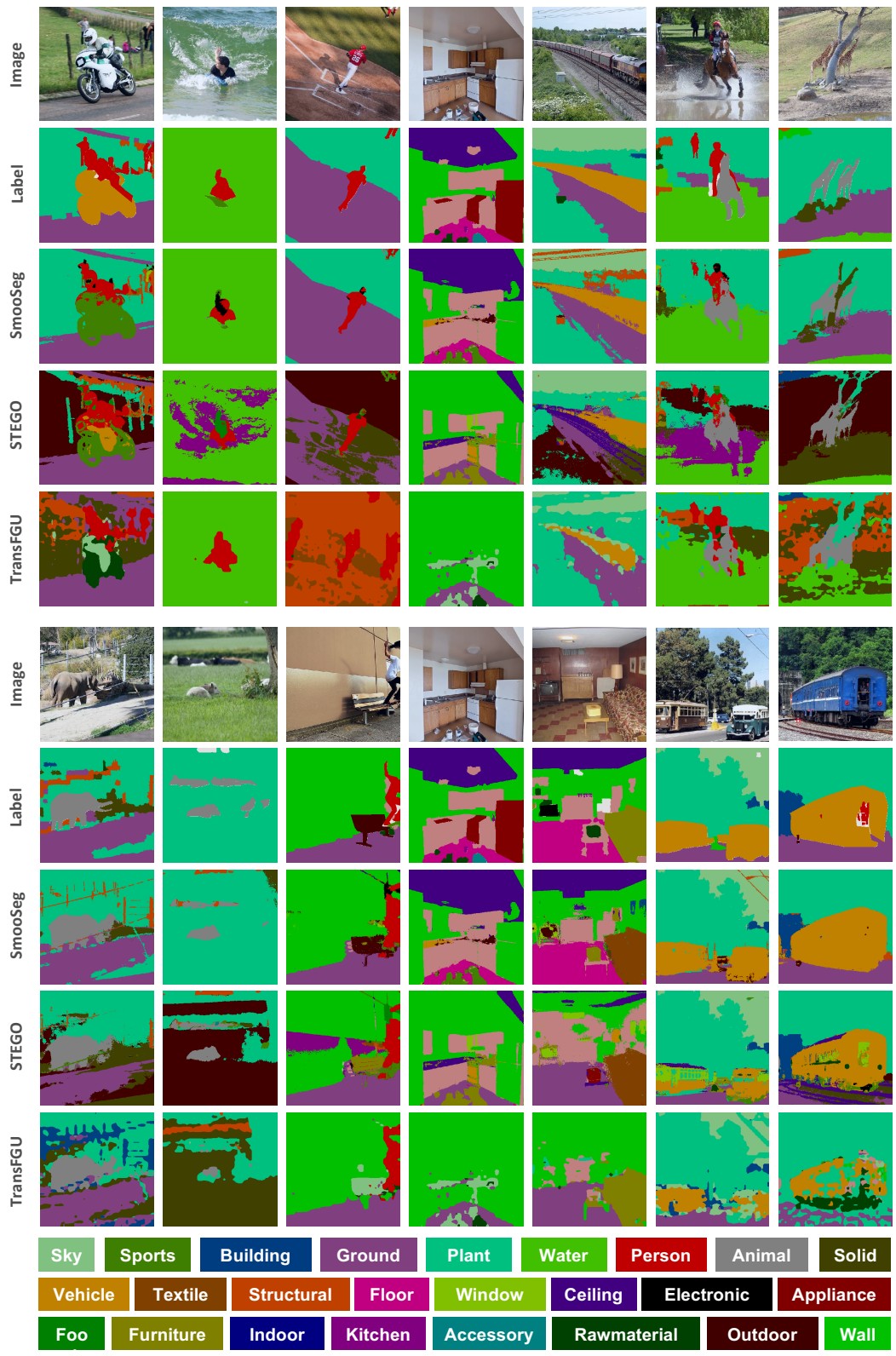

Figure 12: Additional qualitative results on the COCOStuff dataset. It's evident that SmooSeg produces higher-quality segmentation maps compared to STEGO and TransFGU.

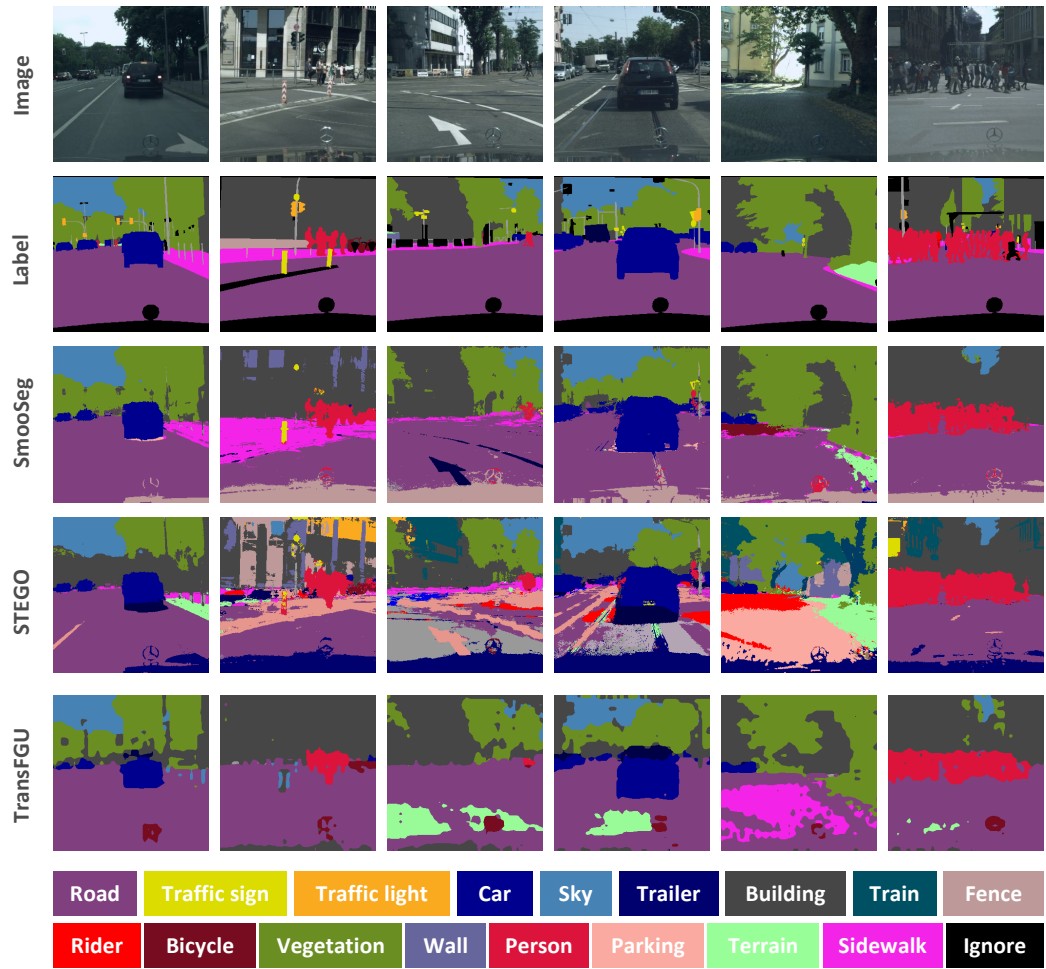

Figure 13: Additional qualitative results on the Cityscapes dataset.

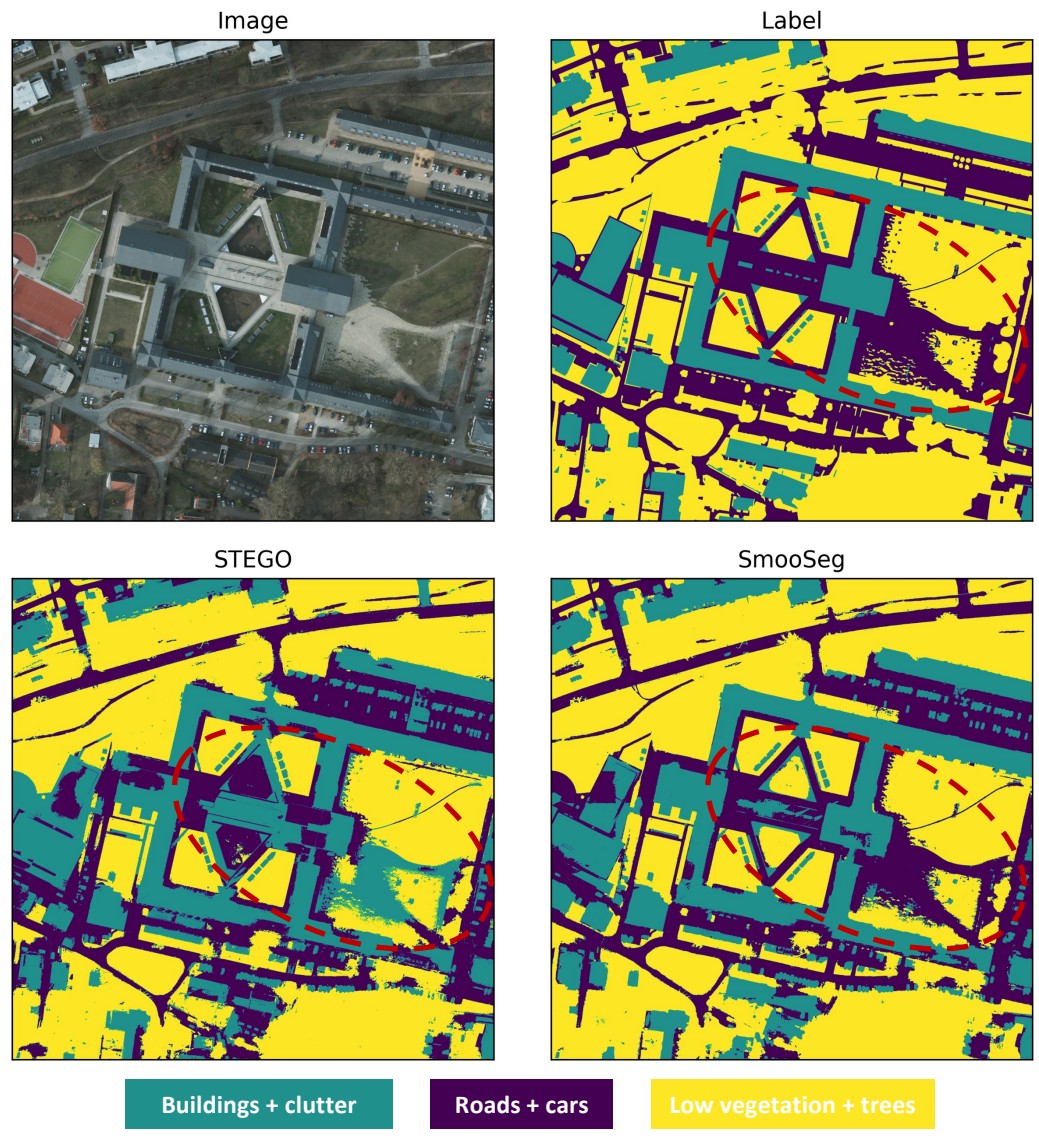

Figure 14: Additional qualitative results on the Potsdam-3 dataset. The large-scale image have a resolution of $4800 \times 4800$, which will be partitioned into 225 sub-images with a resolution of $320 \times 320$ before being input into each algorithm.

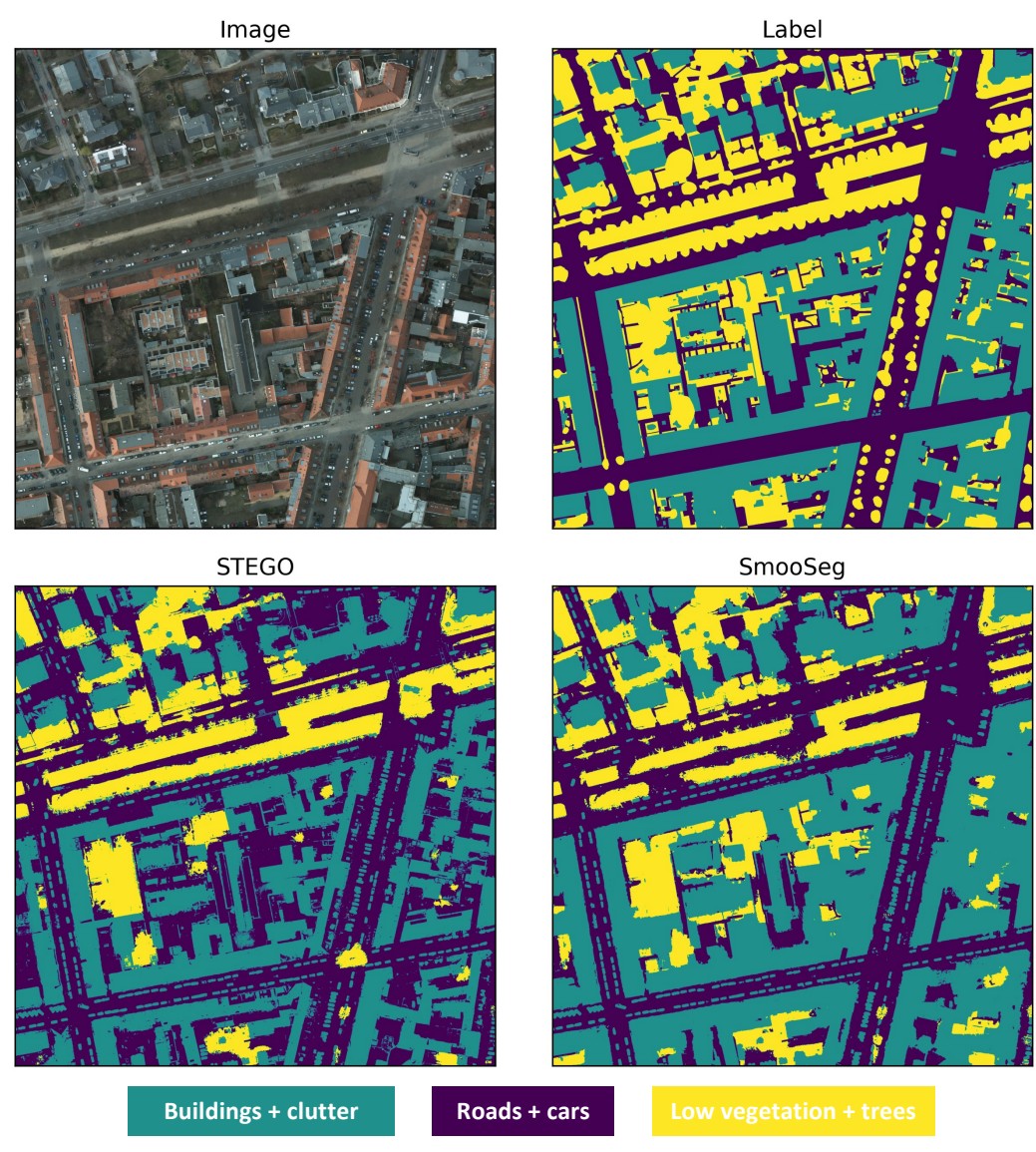

Figure 15: Additional qualitative results on the Potsdam-3 dataset.

