# OpenReview forum: "SmooSeg: Smoothness Prior for Unsupervised Semantic Segmentation"
_NeurIPS.cc/2023/Conference — NeurIPS 2023 poster_

### Official Review · Reviewer_yftt · 2023-06-25

**Soundness:** 3 good
**Presentation:** 3 good
**Contribution:** 2 fair
**Rating:** 4
**Confidence:** 5

**Summary:**

The paper tackles unsupervised semantic segmentation which aims to group pixels into semantic clusters without manual annotation. This paper proposes the smoothness prior by enforcing that adjacent features in metric space share the same semantics. The approach relies on features from a self-supervised representation learning method (ie., DINO). Additionally, the architecture relies on an asymmetric teacher-student style predictor that updates pseudo labels, leading to smoother segmentation predictions. This approach outperforms STEGO (prior s-o-t-a) in terms of pixel accuracy on three datasets: COCOStuff, Cityscapes, and Potsdam with a ViT-S backbone.

**Strengths:**

### Method:
- Leverages advantages in self-supervised learning (SSL): The proposed method leverages the advancements in self-supervised representation learning in generating dense representations. Notably, it proposes to model the relationships between image patches by relying on high-level features from a pre-trained DINO model.
- Kmeans is not required: The introduced smoothness prior can be used to directly optimize the generated segmentation map. The paper claims this strategy leads to more coherent and accurate segmentation results than prior work STEGO. In particular, the latter relies on minibatch Kmeans to obtain the final segmentation map.
- Learnable prior: The approach does not rely on grouping priors as explored in prior works (see below for references). Instead, the approach implicitly uses a learnable grouping prior by relying on the self-supervised representation learning method (i.e., DINO).

### Experiments:
- Dataset variety: The approach is applicable to different datasets. In particular, the experimental validation considers 3 different datasets: COCOstuff, Cityscapes, and Potsdam.
- Quantitative results: the approach outperforms prior art (STEGO) on pixel accuracy when using DINO pre-trained weight on Citysapes and COCOstuff dataset.

### Misc.
- Clarity: The method is well presented by relying on clear figures (e.g., Figure 2). The paper is also easy to follow (for someone who’s familiar with the literature at least).


**Weaknesses:**

### 1. Originality Method - Difference with STEGO / SlotCon:

- The paper can be seen as a combination of STEGO and SlotCon.
The method largely follows the methodology of STEGO. The main SmooSeg loss is similar to the correlation loss in STEGO, which is also mentioned in the paper (see L158).  Overall, I don’t think this is a major issue as I haven’t seen this combination being applied to the task of unsupervised semantic segmentation directly.

- (Minor) I also don’t completely follow the reasoning that the proposed method is superior to Kmeans (L165). While this is indeed an advantage, it’s not clear why Kmeans would necessarily perform worse, as used in STEGO (both STEGO and SmooSeg rely on the features from DINO anyway). Kmeans also has to rely on a distance metric (e.g., cosine distance between features), resulting in similar features/patches being assigned to the same cluster.


### 2. Robustness -  setting hyperparameters:
There are many parameters that require supervision during training:
- CRF weights: The weights of the CRF are determined using the annotated validation set. How can these be determined without annotations? It’s not clear from the supplementary if the weights are dataset-specific. (I assume that https://github.com/lucasb-eyer/pydensecrf was used.)
- Number of classes: The number of classes is not always known a priori. In particular, it’s not clear how much the approach relies on this information. As a result, the introduced setup (section 3) is relatively artificial. In most practical applications, this information won’t be available. As a result, there are important experiments missing, where the sensitivity to this parameter is ablated (ideally comparing the mean and variance over multiple runs with other methods (e.g., STEGO, PiCIE).
- Dataset-specific parameters: Overall, there are many training parameters that differ across datasets and require finetuning (i.e., loss parameters). Ideally, the same set of parameters is applicable to new/unseen images, especially for an ‘unsupervised’ method. Now, it’s hard to judge the robustness of the approach as it relies on finetuned parameters for each dataset. To my knowledge, this is not the case for prior works and baselines (IIC, PiCIE, DeepCluster). Also, TransFGU keeps most of its parameters constant. While this issue is somewhat tackled in the supplementary (see Section B), it’s still not clear if this strategy was actually used to set these parameters during training.

### 3. Scope of the experiments:
- The experiments are limited to results on 27 classes  (COCOStuff and Cityscapes). TransFGU also includes COCO-80 and COCOStuff-171. It would be interesting to see the performance for these setups as well.
- A more practical setting is the semi-supervised setup, especially as the ‘unsupervised’ results are relatively poor and not immediately useful for practical applications. Does the method improve the representations by finetuning the learned representation compared to DINO? Finetuning the complete model or simply training a linear probe on top of the model are interesting experiments [f]. I would also expect that the learned representation can be efficiently fine-tuned with only a few samples [b].

### 4. Quantitative results - different backbones:
I noticed that only a ViT-S is used for COCO-stuff and Cityscapes, while a ViT-B is used for the Potsdam dataset. However, STEGO reports its final results with a ViT-B and outperforms the numbers in this paper (i.e., STEGO obtains 28mIoU on COCOstuff with a ViT-B). So, why is a ViT-B not used for COCO-stuff and Cityscapes as well, to make sure that the observations transfer to stronger backbones? This would furthermore make the main claim of the paper stronger.

### 5. Related work
There are also a few related works missing. Interestingly, some of these rely on priors, such as superpixels, edge, or saliency estimators [d, e, f], which the paper under review does not require. More suggestions can be found below:

[a] Wang et al, Dense Contrastive Learning for Self-Supervised Visual Pre-Training, CVPR.

[b] Wang et al, FreeSOLO: Learning to Segment Objects without Annotations, CVPR.

[c] Ziegler et al, Self-Supervised Learning of Object Parts for Semantic Segmentation CVPR.

[d] Hwang et al, Segsort: Segmentation by discriminative sorting of segments CVPR.

[e] VanGansbeke et. al, Unsupervised Semantic Segmentation by Contrasting object mask proposals, ICCV.

[f] Zhang et al., Self-Supervised Visual Representation Learning from Hierarchical Grouping, NeurIPS.


**Questions:**

My most important questions are listed above.

I also have a few additional questions:
- L265 claims that the proposed method outperforms STEGO at the boundaries. However, are the boundaries not primarily dependent on the CRF? How do these methods compare when we don’t use a CRF? It would also be useful to see a few examples without a CRF in the supplementary (optional).
- How is the best model selected in the absence of a validation set? Can the loss function be used to select the best model during training?
- The method is dependent on the pretrained DINO weights. We see that MoCov2 performs worse in Table 1. What happens if we use other weights (MoCov3, Pixpro, DenseCL etc.)?
- Are the visualized predictions in the supplementary randomly selected?


**Limitations:**

While some of the limitations are mentioned in the paper (see Section 4), I suggest including other limitations (e.g., known CRF weights, and known number of classes). I also couldn't find a paragraph on the societal impact.

---

> ### Author Rebuttal · Authors · 2023-08-09
>
> Thank you for taking the time to review our work.
>
> ### Why SmooSeg is superior to Kmeans used in STEGO
> The performance of Kmeans heavily relies on the quality and discriminative power of the features it operates on. In the case of DINO features, as shown in the 2nd column of Figure 5, the Kmeans centers do not possess much semantic meaning as they almost stand at the same point. This phenomenon indicates that the DINO features lack sufficiently discriminative information to learn distinct semantic centers through Kmeans. Additionally, STEGO aims to distill DINO features into a more compact and structured representation using a distillation loss, making the class centers of Kmeans in this learned embedding space more semantically meaningful (3rd column of Figure 5). However, it is worth noting that although the DINO features and STEGO embeddings may exhibit many compact sub-clusters, they are not continuous or coherent in the semantic sense. In contrast, our SmooSeg, empowered by the smoothness prior, can produce highly semantically compact and coherent semantic clusters, as evidenced by the last column of Figure 5.
>
> ### Setting hyperparameters
> - **CRF weights:**
> Please see the general response.
>
> - **Number of classes:**
> Predefining the number of classes appears to be a common practice in existing studies that address the problem of unsupervised semantic segmentation. PiCIE adopts the number of classes as the number of cluster centroids in its algorithm; TransFGU adopts this number as the number of class embeddings in its Segmenter module; and STEGO requires a predefined number of classes when performing Kmeans. We follow this problem setting in our work for fair comparison but agree that this deserves further studies.
>
> - **Dataset-specific parameters:**
> We agree that tuning parameters without a validation set can be challenging for unsupervised methods. While some prior works (e.g., IIC and PiCIE) do not explicitly have dataset-specific parameters, they generally perform significantly worse than those with (e.g., TransFGU and STEGO). In view of the parameter setting challenge, we strive to find a practical solution, which leads to the analysis in Appendix B. Through experimentation, we found that stable results can be achieved when b1 is set at 0.5 and b2 is nearby 0.0, which may to some extent enhance the practicality and robustness of SmooSeg in real-world applications.
>
> ### Scope of the experiments
> - **Additional experiment**
> Thanks for your advice. COCOStuff-171 and COCOStuff-80 require much more computations and we are not able to obtain their results before the rebuttal deadline. We will add their results in the revision.
>
> - **Semi-supervised setup and few-shot setup**
> We appreciate the reviewer for providing these constructive suggestions. As can be observed in Figure 5, the embedding of SmooSeg exhibits significantly higher semantic consistency compared to the features of DINO, which confirms that SmooSeg is able to enhance representations over DINO. We are also curious if the representations learnt by SmooSeg can be further fine-tuned with a few samples. As the current work focuses on the unsupervised setting, we would like to reserve this exploration to future studies.
>
> ### The choice of ViT-S
> The choice of using ViT-S as the backbone on the COCO-stuff and Cityscapes datasets was made to ensure a fair comparison with both STEGO and TransFGU, as TransFGU only adopts ViT-S as its backbone. Accordingly, the results of TransFGU are directly cited from their original paper.
>
> ### Missing references
> We appreciate the reviewer for providing these valuable references. In our revision, we will include a thorough discussion with these literatures to highlight the distinctions in approaches.
>
> ### The performance and examples without CRF
> Please see the general response.
>
> ### Select the best model without a validation set
> In practice, the loss from the pretext task may not always be a reliable indicator of the model's performance on actual downstream tasks. Basically, it is impossible to select the “best” model, but there are some potential approaches to choose a good model without a validation set:
> - Setting a global training step and selecting the last saved mode. As the training progresses, the model tends to become more stable, and the performance may also stabilize.  In our experiments, we have found that the checkpoint at the end of training often shows good results.
> - Making visualizations of segmentation maps for some training samples during training. By comparing these results, one can qualitatively assess the performance of the model and select a model that shows promising segmentation results.
>
> ### The performance with different backbones
> Following you suggestion, we further conduct experiments on COCOStuff by using Pixpro and DenseCL weights for SmooSeg.
>
> | Methods            | Acc  | mIoU |
> | --------           | ---- | ---- |
> | SmooSeg + DINO     | 63.2 | 26.7 |
> | SmooSeg + MoCoV2   | 52.4 | 18.8 |
> | SmooSeg + pixpro   | 48.3 | 18.1 |
> | SmooSeg + DenseCL  | 54.6 | 19.2 |
>
> DINO features indeed yield the best performance of SmooSeg when compared to its partnership with other SSL representations. Its superority on dense prediction tasks is also evident from its widespread adoption across the literature (in STEGO, TransFGU, and Deep spectral methods [12]). On the other hand, the ResNet architectures employed in models like Pixpro, DenseCL, and MoCov2 have certain limitations in generating low-resolution coarse features, rendering its challenges used for fine-grained semantic map prediction, especially under unsupervised setup.
>
> ### How to select the visualized predictions
> To ensure a comprehensive representation of the model's performance, we manually selected a diverse and representative set of images for visualization. To provide further evaluation, we have included some difficult samples predicted by our SmooSeg in the attached PDF file.

---

> > ### Comment · Reviewer_yftt · 2023-08-13
> > **Response to authors after the rebuttal**
> >
> > I thank the authors for providing the rebuttal. There are several points I want to emphasize after reading the rebuttal and the other reviews:
> >
> > 1. The motivation why the approach outperforms STEGO could be further corroborated by also including "linear probing" results. Why is this information not provided in the paper? STEGO also includes this.
> >
> > 2. One of the main weaknesses is that the approach requires dataset-specific hyperparameters. This limits the scalability of the approach and furthermore contradicts the claim that the approach is unsupervised. While STEGO has shown better results by changing the hyperparameters across datasets, it’s reasonable to argue that this is far from realistic. Additional experiments are necessary to quantify the robustness of the approach as the adopted setup can be considered artificial: the exact amount of clusters, supervised CRF weights, and ideal loss weights are currently being used. As a result, including linear probing and over-clustering experiments would make the paper certainly stronger.
> >
> > 3. In addition, semi-supervised results (e.g., with a linear probe or a shallow head) would also make the approach more useful in practice as the current mIoU scores are relatively low (e.g., 18 mIoU on Cityscapes).
> >
> > 4. I also still don’t understand why different architectures are being used. As the approach heavily relies on STEGO, it makes more sense to include results with a ViT-B for Cityscapes. This is especially important since STEGO reports higher numbers than currently presented in the paper. This would confirm the claims in the paper.

---

> > > ### Author Response · Authors · 2023-08-15
> > >
> > > Thank you for the prompt response.
> > >
> > > **1) Regarding the linear probing:** We would like to clarify that linear probing is a **supervised** approach for assessing the quality of the representations generated by self-supervised representation learning methods. Therefore, we think that linear probing is not suitable for evaluating the performance of SmooSeg as labels are not accessible in unsupervised semantic segmentation. We would like to further elaborate on the rationale as below.
> > >
> > > Existing unsupervised semantic segmentation methods can be roughly categorized into two types:
> > > - **One stage:** IIC[3], PICIE[4], HSG[10], TransFGU[6], and the proposed SmooSeg, which directly learn semantic labels, with losses typically defined on labels.
> > > - **Two stages:** STEGO[8], representation learning + Kmeans for unsupervised semantic segmentation.  In the representation learning stage, losses are defined on feature maps.
> > >
> > > For one-stage methods, as they directly output the predicted semantic maps instead of intermediate features, their performance is evaluated by Acc and mIoU, which are task-specific metrics for unsupervised semantic segmentation. Linear probing is thus not directly applicable and not reported in one-stage methods [3, 4, 6, 10], as the quality of intermediate features is not the objective of these studies. For the two-stage method, STEGO, as it outputs good representations to be used by K-means for images, it is thus reasonable to assess the quality of the learnt representations by linear probing. Therefore, it is not sensible and unfair to compare the results of linear probing between a one-stage and a two-stage method due to their distinct objectives.
> > >
> > > In addition, linear probing may not be a reliable assessment even for two-stage unsupervised segmentation methods because of its sensitivity to feature dimensionality. To illustrate this point, we cite the **linear probing results** from Table 2, page 5, in [a]:
> > >
> > > | Method   | COCOStuff   | Cityscapes |
> > > | -------- | --------    | --------   |
> > > |          | Acc / mIoU  | Acc / mIoU |
> > > | DINO (ViT-B)| 75.8 / **44.4** | **91.3** / **34.9** |
> > > | STEGO    | **76.1** / 41.0 | 89.6 / 28.0 |
> > >
> > > It is shown that the linear probing results of DINO (dimensionality of features: 768) are better than STEGO (dimensionality of embedding: 90 or 100). These linear probing results contrast with the superiority of STEGO over DINO for the unsupervised semantic segmentation task.
> > >
> > > *[a] Uncovering the Inner Workings of STEGO for Safe Unsupervised Semantic Segmentation. CVPR, 2023.*
> > >
> > >
> > > **2) Semi-supervised setup.** Our method aims to address the problem of unsupervised semantic segmentation rather than semi-supervised semantic segmentation [b,c].
> > >
> > > *[b] Semi-supervised semantic segmentation with prototype-based consistency regularization, NeurIPS 2022.*
> > >
> > > *[c] Semi-supervised semantic segmentation via gentle teaching assistant, NeurIPS 2022.*
> > >
> > > **3) Problem setup and hyperparameter issue.** We strictly follow the established unsupervised semantic segmentation setup as in [3,4,6,8], where **the number of clusters is predefined**. For example, this number is used as the number of cluster centroids in PiCIE [4], the number of class embeddings in TransFGU [6], and the "K" in Kmeans in STEGO [8]. In addition, it's important to highlight that **CRF with default settings is employed only during the testing phase**. Therefore, its parameters are not relevant to the hyperparameters of our method. Finally, most existing SOTA methods, including STEGO and TransFGU, also contain dataset-specific hyperparameters.
> > >
> > > To further demonstrate the effect of the number of classes, we tune the number of prototypes in SmooSeg and show the results below. The best result of SmooSeg is obtained when the number of prototypes equals the ground-truth number of classes. Interestingly, even with more prototypes, SmooSeg consistently outperforms STEGO (Acc: 48.3, mIoU: 24.5).
> > >
> > > | Number of prototypes | 27                  | 30          | 33          | 37          |
> > > |:--------------------:|:-------------------:|:-----------:|:-----------:|:-----------:|
> > > |                      | Acc / mIoU          | Acc / mIoU  | Acc / mIoU  | Acc / mIoU  |
> > > | SmooSeg              | **63.2** / **26.7** | 58.5 / 25.6 | 61.3 / 25.3 | 59.1 / 24.2 |
> > >
> > >
> > > **4) Additional results with DINO ViT-B/8 on the Cityscapes dataset.** The results of SmooSeg with DINO ViT-B/8 as the backbone are provided below. It's evident that SmooSeg is also superior to STEGO under this backbone.
> > >
> > > | Method           |    Acc      |   mIoU    |
> > > |       --------   |   --------  |  -------- |
> > > | STEGO (ViT-B)    |    73.2     |   21.0    |
> > > | SmooSeg  (ViT-B) |    **84.5**     |   **21.5**   |

---

> > > > ### Comment · Reviewer_yftt · 2023-08-16
> > > > **Response**
> > > >
> > > > Thank you for the discussion. I appreciate the points.
> > > >
> > > > - Linear probing: I partially disagree/agree. Indeed, linear probing is an excellent tool to measure the quality of the obtained representations. However, it is still extremely unlikely that the exact number of classes is known a priori. Furthermore, this single linear layer can quickly adapt when only a few labels are available.  The adopted Hungarian matching algorithm is somewhat 'flawed'. In particular, this 'perfect' matching criterion (during evaluation) also assigns probability mass to classes for which the model might have predicted close to 0%. As a result, I still encourage the authors to add linear probing results to the paper as done in many prior works, such as in STEGO, HSG, and most of the related work mentioned in my initial review [C, E, F].
> > > >
> > > > - Hyperparameters: The issue is not that K is predefined, but that we don't have any information on the robustness of these parameters when compared to STEGO or other prior approaches. The presented approach seems to benefit from knowing the exact classes (see table above) and furthermore relies on dataset-specific parameters. While 1 or 2 works seem to adopt this setting, most prior works don't.  As a result, adding over-clustering results to the paper and also keeping the hyperparameters fixed for similar datasets/domains, is important for an 'unsupervised' method. For all we know, STEGO might be more robust to these parameters which would benefit scaling.
> > > >
> > > > - Semi-supervised results: I understand that semi-supervised learning is not the aim of the paper. My point is that the current mIOU scores are low. Adding semi-supervised results as in [B] would make the submission stronger.
> > > >
> > > > All in all, I'm questioning the usefulness of the approach as it requires the classes to be known a priori, as it relies on dataset-specific hyperparameters, and as it performs worse in the presence of (a few) labels than the baseline (DINO). Based on these points I'm still leaning towards a borderline score.

---

> > > > > ### Author Response · Authors · 2023-08-18
> > > > > **Response to Reviwer yftt**
> > > > >
> > > > > Thank you for the response.
> > > > >
> > > > > **1) Linear probing.**
> > > > >
> > > > > - To the best of our knowledge, DINO [11], STEGO [8] and the works [C, E, F] are self-supervised representation learning methods (SSL). When applied to the task of unsupervised semantic segmentation (USS), they belong to the category of two-stage methods. As elaborated in our previous discussion, it is reasonable for them to assess **their representation learning quality** by linear probing. In contrast, one-stage methods for USS such as IIC [3], PiCIE [4], TransFGU [6], and HSG [10] aim to directly predict the final semantic maps with losses imposed to the labels rather than to features. That is why these methods **did not report the quality of their intermediate features by linear probing as obtaining good feature representations is NOT their goal**. Our SmooSeg is a one-stage method for USS and not an SSL method; thus we don’t see a need to discard our predictor dedicated for USS and assess the quality of intermediate features via linear probing as what SSL methods do.
> > > > >
> > > > > **2) Hyperparameters.**
> > > > > - **The number of classes.** Setting the number of classes as a hyperparameter is typical for unsupervised learning, which can be traced all the way back to kmeans, a classic unsupervised learning method. **We certainly cannot conclude that kmeans is not an unsupervised method due to the use of k**. The same applies to the solutions to USS: as long as the proposed methods do not utilize any annotation (label information), they are unsupervised methods. In fact, when applying STEGO to USS, it also requires to input a value of k to kmeans and use Hungarian matching for class alignment. If one applies STEGO to USS by linear probing, it is not called “unsupervised” due to the use of labels.
> > > > >
> > > > >      To clarify your concerns, we adjust the parameter "K" of Kmeans in the STEGO and present the outcomes below. We use the checkpoint of STEGO (ViT-B) on the COCOStuff dataset provided by the authors. It can be seen that the performance of STEGO also diminishes under the overclustering setting.
> > > > >
> > > > >     | K | 27                  | 30          | 33          | 37          |
> > > > >     |:--------------------:|:-------------------:|:-----------:|:-----------:|:-----------:|
> > > > >     |                      | Acc / mIoU          | Acc / mIoU  | Acc / mIoU  | Acc / mIoU  |
> > > > >     | STEGO (ViT-B)              | **56.9** / **28.2** | 53.5 / 23.3 | 52.7 / 23.7 | 51.9 / 20.1 |
> > > > >
> > > > > - **Dataset-specific parameters.** We do not agree that STEGO is more robust to dataset-specific parameters. Not including k needed in kmeans, **STEGO has 6 dataset-specific parameters**. It is shown in Table 6 of STEGO’s Appendix A.10, these parameters can vary significantly across different datasets (e.g., $\lambda_{rand}$ is 0.91 on Cityscapes and 0.15 on CocoStuff). In fact, we have spent a lot time tuning these hyperparameters of STEGO when obtaining its results on ViT-S as its performance is sensitive to them. On the contrary, **our SmooSeg has 2 dataset-specific parameters**, which are observed to be more stable across different datasets (Table 5 in our supplementary). We have strived to find a practical way to set their values in our Appendix B.
> > > > >
> > > > > **3) Semi-supervised results.**
> > > > > - We acknowledge that there is a certain gap between unsupervised and semi-supervised semantic segmentation. However, it’s important to underscore that our SmooSeg demonstrates clear superiority over existing USS methods, thus effectively establishing its efficacy. In addition, it’s essential to recognize that USS methods do not solely pursue an absolute segmentation outcome; rather, their primary objective lies in showcasing higher performance within the unsupervised setting without any annotation.

---

### Official Review · Reviewer_iJZU · 2023-07-03

**Soundness:** 3 good
**Presentation:** 4 excellent
**Contribution:** 2 fair
**Rating:** 6
**Confidence:** 4

**Summary:**

The paper presents an unsupervised training method for semantic segmentation models. It trains the network on a loss with two terms: one encouraging smoothness in the segmentation labels, and a "data" term based on self-training. It assumes a neural net architecture with a) an extractor that produces per-pixel features $X$, b) a projector that produces lower-dimensional features, to be compared against "prototypes" and used in self-training, and c) the "predictor" that outputs the per-pixel segmentation labels from the prototype/feature comparisons. The backbone feature extractors are primarily based on DINO [11], though some experiments appear to use MoCo [24] in its place. This is evaluated in terms of its accuracy in semantic segmentation tasks including COCOStuff, Cityscapes, and Potsdam-3.

**Strengths:**

i) Experiments consider variety of semantic segmentation tasks: street, general, and aerial image datasets.

This is a good demonstration that the method can be applied generally, even if it is sensitive to some choice of hyperparameters.

ii) Uses a backbone (DINO) also trained in an unsupervised (self-supervised) way.

Uses DINO ViT-S/8 [11] or MoCo [20] as backbones in different experiments on the segmentation head & training. These were also trained without hand labelling, wtih a self- or unsupervised method; as opposed to using, for example, a standard ImageNet backbone from supervised training. This is a correct and principled way to make sure the proposed segmentation models are truly and completely unsupervised.

iii) Precise presentation

Uses both an algorithm, written in plain code, along with diagrams and the math to describe the method quite completely. Those parts of the method that are unclear from the definition of the loss can be deduced quite unambiguously from Algorithm 1, and vice versa.

**Weaknesses:**

iv) Not a lot of experiments or analyses explaining *why* the method works.

The experiments mostly show end results (mAP, qualitative segmentation quality). Hyperparameter studies are useful, though: providing good insight into the role each hyperparameter plays. And there is a minimal ablation study in Table 4.

It still seems unclear what exactly the network is learning from. For example, to what extent is the smoothness a constraint on the extractor vs. the projector? It seems possible that $E_\mathrm{smooth}$ is also penalizing the cosine distances, as defined in Equation (2), and pushing feature vectors $X$ together, and not just using it as a weight for spatial smoothness in the labels $Y$, as in the intuitive explanation of Eq (1). I don't see `torch.no_grad` or any equivalent around the weight computations in Algorithm 1. Fig 5 visualizes feature embeddings on Potsdam-3, and this shows that the group similar classes together. Though, if the mapping between feature vectors $X$ and the labels $Y$, in the projector and predictor, is a "smooth" function (in the sense of Lipschitz smoothness) so that similar features get similar labels, might we already expect that the smoothness penalty is reduced, even if there is no meaningful spatial smoothness of the labels?

The relative role/contribution of the within-image and between-image comparisons also seems to complicate the intuitive explanation of the method. There is a large accuracy drop if $E_\mathrm{smooth}$ only penalizes adjacent pixels within the image, and not across the images. The motivation for $E_\mathrm{smooth}$ that adjacent pixels are likely to have the same label, the "natural tendency towards piecewise coherence regarding semantics, texture, or color," does not apply to pixels within different images. The smoothness between pixels that are adjacent in a metric space is a property of the construction of that metric space: not a natural property of images, so this part's role in the function of the smoothness perhaps needs more explanation and experiments. Why do we expect with this unsupervised loss that the metric space will be semantically meaningful?

v) Novelty is arguably more limited than is claimed.

The use, in unsupervised training of semantic segmentation models, of spatial smoothness of the labels does seem to be explored. Though many of the particulars and how it's used within the overall framework is new as of this submission.

Smoothness priors themselves are, of course, quite common in semantic segmentation. Spectral methods also expected to be a form of smoothness prior, but using squared differences in place of the $\delta$ in Eq (1) of the submission. This kind of prior is used in unsupervised training of semantic segmentation models in:

[a] Xia & Kulis "W-Net: A Deep Model for Fully Unsupervised Image Segmentation"

[b] Melas-Kyriazi et. al. "Deep Spectral Methods: A Surprisingly Strong Baseline for Unsupervised Semantic Segmentation and Localization"

W-Net also incorporates a fully-connected CRF, in section 4.1, though this is only for preprocessing and not for unsupervised training. I however don't see prior work that uses a fully-connected CRF within the training objective in the way that this submission does. The backbone is also updated relative to previous work: this submission is perhaps in the genre of papers that applies a given vision idea/concept to ViTs instead of CNNs.

vi) There may also be some gaps in the citation of prior work similar to the submission's teacher/student/self-training approach described in section 3.2. Citations on self-supervised learning in the "Related Work" in section 2 focus on contrastive learning approaches. More similar approaches might include:

[c] Scudder "Probability of error of some adaptive pattern-recognition machines"

[d] Xi et. al. "Self-training with Noisy Student improves ImageNet classification"

Based on my read, the submission does *not* modify the inputs or consider different views between teacher and student, as in [19, 20, 21, etc.]. (If my read is incorrect, what exactly is the "pretext task" that the authors use?) Though [c.d] and similar papers differ enough from the submission's training scheme, including the use of a projector module to get the prototypes that are compared between teacher & student, that this part is not relevant to judging novelty.

**Questions:**

Why the focus on semantic segmentation? If the labels aren't used, then I'd assume the method should equally apply to instance, panoptic, or salient-foreground segmentation. Especially given it does seem to generalize across different kinds of semantic segmentation. Is it because the prototypes $P$ are expected to somehow relate to semantic classes of "stuff?"

The stated goal of the smoothness prior is that "the segmentation model is encouraged to assign similar labels to adjacent patches, thereby promoting spatial coherence within objects." What determines which pixels are "adjacent" when constructing E_smooth? I.e. is it 4-connected, 8-connected, or fully connected? The pairwise weights don't seem to include spatial distance. Toward the end of Section 3.1 it's stated that the authors "also apply the smoothness prior across images." Which pairs of pixels across pixels are included as terms in the sum in Eq (1)?

Minor comments:

  * Typos in Algorithm 1 comments: "updata" and "prototyeps"
  * Citation to [20] is repeated on line 94

**Limitations:**

The sensitivity of the results to hyperparameter choices is mentioned as a limitation. This does seem likely to be the major drawback, with bad results like mode collapse being a concern. In Appendix B of the supplement, it is described in qualitative terms that one can monitor the distribution of the differences in soft label assignments (the "smoothness degree") to adjust the parameters. They don't give a fully reproducible method or algorithm for determining b1 and b2 to achieve the desired $\delta$ distribution, but it does seem clear what one would be looking for to tune this manually.

---

> ### Author Rebuttal · Authors · 2023-08-09
>
> Thank you for taking the time to review our work.
>
> ### Why the method works
> **What the network is learning from.** Please refer to the general response.
>
> ### The relative role/contribution of the within-image and between-image comparisons.
> We apologize for any confusion caused. We clarify that our “adjacent” patches are defined in the high-level feature space generated by the frozen pre-trained model, rather than in the spatial coordinate space of images. That is, adjacent patches in our model refer to patches with similar extracted features: they may or may not lie closely in a single image and may belong to different images. Your insight regarding the performance drop when penalizing only adjacent pixels within an image aligns with our observations. The rationale behind this phenomenon lies in two folds: 1) the smoothness term across the images could provide the negative force that prevents the model collapse; 2) the smoothness term across the images helps construct a global and semantically meaningful distribution across the entirely dataset. While it is intuitive that similar patches within the same image should share the same label, the smoothness term across images extends this concept to all patches with the same semantics across all images. This promotes the creation of coherent and meaningful semantic clusters that transcend individual images, as shown in Figure 5 in the main submission.
>
> We argue that the concept of smoothness between image patches in a metric space is reflective of the natural properties of images. This assertion is grounded on the fact that a pre-trained model can extract semantically consistent features from images. For instance, patches belonging to a common object like a dog would naturally cluster together both in the pixel space and in the metric space when provided with semantically consistent features. This highlights the reason behind our decision to construct the closeness matrices in the metric space, as opposed to the spatial coordinate space.
>
> ### Difference with W-Net and Deep Spectral Methods
> We deeply appreciate the reviewer's kind provision of references and recognition of the originality inherent in our overall framework. While acknowledging the utilization of a fully-connected CRF in W-Net for preprocessing, it's crucial to emphasize that our submission distinguishes itself by incorporating a smoothness loss within the context of unsupervised training, rather than being limited to preprocessing or postprocessing stages. Furthermore, we'd like to clarify that Deep Spectral Methods do not encompass model training; they utilize spectral decomposition for object or foreground segmentation. It's worth noting that Deep Spectral Methods lack the capability of distinguishing the "stuff" category. Moreover, we concur with your observation regarding the updated backbone: our method aligns particularly well with ViTs structure.
>
> ### Missing citation of prior work
> We sincerely appreciate the reviewer for pointing out the missing citations. In our revision, we will ensure to include these relevant works and provide a more comprehensive discussion on similar approaches. We also appreciate the reviewer's accurate understanding of our work: SmooSeg does not modify the inputs or consider different views between teacher and student.
>
> ### Why focusing on semantic segmentation
> The focus on semantic segmentation in SmooSeg is mainly due to the way prototypes are learned. In SmooSeg, the prototypes are automatically learned from image patches without any direct constraints, making it challenging to differentiate between foreground and background prototypes or distinguish different instances.
>
> Indeed, smoothness has the potential to be incorporated into various types of semantic segmentation as a smoothness regularization term. However, for tasks requiring finer distinctions, such as instance segmentation or salient foreground segmentation, additional modifications and research may be needed to address the specific challenges associated with those tasks. Its extension to other types of segmentation tasks is an interesting direction for future research.
>
> ### The construction of closeness matrices $W^{i,i}$ and $W^{i,i^{\prime}}$
> While early smoothness-prior-based methods often define adjacent pixels for a given pixel based on the 4-connected or 8-connected grid in the coordinate space, we argue that this approach neglects the high-level semantic information of image patches. In contrast, SmooSeg defines the closeness relationship among image patches in a metric space (the feature space of self-supervised representation learning methods) rather than in the coordinate space, by calculating the cosine similarity of high-level features. The resulting closeness matrix is fully connected, representing the adjacency between all image patch pairs. Theoretically, a large element value in the closeness matrix, indicating a high cosine similarity, suggests a high possibility of "adjacent" patch pair, and vice versa. From this perspective, when minimizing the smoothness loss, SmooSeg encourages two patches to have similar labels if their relationship in the closeness matrix is positive, and vice versa. SmooSeg enables the utilization of high-level semantic information to guide the smoothness regularization, leading to improved segmentation results that account for semantic coherence and consistency between image patches.
>
> We apologize for the confusion on the smoothness prior across images. In our smoothness term given by Eq (4), there are two closeness matrices involved—one for within an image and another for across images. Specifically, the closeness matrix across two different images $I_i$ and $I_{i^{\prime}}$ is computed as the cosine similarities of all across-image patch pairs:
> $W_{pq}^{ii^{\prime}}=\frac{X_{i,p} \cdot X_{i^{\prime}, q}}{||X_{i, p}|| \ ||X_{i^{\prime}, q}||}.$
>
> ### Typos and repeated citation
> We will proofread the paper again and correct all the typos.

---

> > ### Comment · Reviewer_iJZU · 2023-08-21
> >
> > >  We clarify that our “adjacent” patches are defined in the high-level feature space generated by the frozen pre-trained model, rather than in the spatial coordinate space of images
> >
> > This is a good clarification, thanks! Definitely a few passages I'd suggest editing in the camera-ready keep this clear. For instance, the intro seems to be setting up to motivate smoothness in the image space, or conflate the two:
> >
> > > However, despite their effectiveness, these methods often overlook the property of spatial coherence of image segments
> >
> > > Observations close to each other, either in the form of neighboring pixels or adjacent features in a metric space, are expected to share similar semantic labels
> >
> > or, in section 3,1:
> >
> > > In other words, the segmentation model is encouraged to assign similar labels to adjacent patches, thereby promoting spatial coherence within objects.

---

> > > ### Author Response · Authors · 2023-08-21
> > > **Thanks for your valuable suggestions**
> > >
> > > Thanks a lot for your valuable suggestions that help improve our paper. We will certainly address this confusion in our revision. Please let us know if there are any additional concerns or suggestions that could further enhance the quality of this work.

---

### Official Review · Reviewer_gyDm · 2023-07-03

**Soundness:** 3 good
**Presentation:** 3 good
**Contribution:** 3 good
**Rating:** 6
**Confidence:** 4

**Summary:**

This work addresses the problem of unsupervised semantic segmentation. In contrast to STEGO (baseline), the approach learns semantic embeddings directly in a student-teacher regime without the need for the K-means. The objective is entirely unsupervised and is reminiscent of a CRF energy formulation with data and a smoothness term. The pseudo labels from the teacher embeddings provide the signal for the data term. The work achieves improved segmentation accuracy on standard benchmarks over the state of the art and offers an interesting analysis. Overall, the work is compelling in quality and contribution.

**Strengths:**

- The application of the smoothness constraint is natural and simple.
- The learning problem is well-designed and executed. The text is generally well-written (albeit not without minor typos).
- The experiments have sufficient scope and provide valuable insights. The empirical results are compelling.

**Weaknesses:**

There are technical similarities to STEGO, such as in the loss formulation in Eq. (4). There are differences, though, as discussed in 158-174. However, it does take away a bit from the novelty.

Empirically, the approach appears at its best with DINO features using ViT architecture; the improvement over prior art may not translate to other SSL representations and model architectures (MoCo, ResNet, see Tab. 1). Nevertheless, the approach remains competitive.

The method introduces a number of hyperparameters, which can be challenging to fine-tune in an unsupervised setup. I like that this is acknowledged in the work, however, and Appendix B provides some interesting analysis.

Typos:
e.g. l. 178, 307

**Questions:**

- l. 140-141: Why is optimisation only stable with such normalisation?
- Eq. 5: Why is there a need to stop the gradient flow in this fashion?
- l. 212: I would be curious to understand how the predictions and the ground truth are aligned in a bit more detail.
- How were the other parameters (e.g. the momentum, the temperature) chosen/fine-tuned?
- I do not quite follow the sentence in l. 172: “which represents discontinuities between image patches that should be preserved.”

**Limitations:**

The limitations are discussed in the main text. The supp. material provides further analysis.

---

> ### Author Rebuttal · Authors · 2023-08-09
>
> ### Technical similarities to STEGO
> We acknowledge some technical similarities between our smoothness loss and the correlation loss in STEGO. These loss formulations are not entirely novel and have been seen in various dimensionality reduction methods (e.g., [a]). Essentially, STEGO boils down to an unsupervised dimensionality reduction method [b], followed by a kmeans grouping of learned embeddings for patches. In contrast, our method goes beyond dimensionality reduction and utilizes the smoothness prior to facilitate the learning of the labeling function, which encourages piecewise smoothness and leads to more coherent and semantically meaningful semnentation maps. Additionally, our smoothness loss directly constrains the label assignment, which brings a desirable property when compared to the correlation loss that operates on the reduced feature correspondence.
>
> *[a] He, Xiaofei, and Partha Niyogi. "Locality preserving projections." Advances in neural information processing systems 16 (2003).*
>
> *[b] Koenig, Alexander, Maximilian Schambach, and Johannes Otterbach. "Uncovering the Inner Workings of STEGO for Safe Unsupervised Semantic Segmentation." In Proceedings of the IEEE/CVF Conference on Computer Vision and Pattern Recognition, pp. 3788-3797. 2023.*
>
>
> ### Backbone choice
> DINO features using ViT architecture indeed yield significant benefits for downstream dense prediction tasks when compared to other SSL representations. This is evident from its wide adoption in the literature, including STEGO, TransFGU, and Deep spectral methods [12].
>
> ### Hyper-parameter choice
> Fine-tuning hyper-parameters is a common requirement in state-of-the-art unsupervised methods, including STEGO and TransFGU. Therefore, we strive to find a practical solution to address this challenge, which leads to the analysis in Appendix B.
>
> ### Why is optimisation only stable with such normalisation
> By treating the closeness matrix as a weight matrix between nodes in a graph, the zero-mean normalization balances the negative and positive forces during optimization. This balance ensures that the optimization process is more stable, preventing excessive influence from either the negative or positive components of the closeness matrix. Consequently, this normalization contributes to a smoother and more consistent learning process. We will clarify this in the revision.
>
>
> ### Why is there a need to stop the gradient flow in this fashion
> The stop-gradient operation in Eq. 5 is an essential step in our asymmetric student-teacher style predictor. In our approach, there is no observed semantic map available for the data term, so we adopt a self-training strategy to minimize the data loss. The self-training relies on the teacher branch to generate enhanced pseudo labels, which are then used to supervise the learning of the student prototypes. To ensure stability and prevent rapid updates during each training batch, the stop-gradient operation is necessary. On the other hand, we stop the gradient flow from the data loss to the projector’s parameters. This decoupling mechanism for the projector and the predictor will make the learning easier and more stable, as we can avoid the need to trade-off between these two losses during model training.
>
> ### how the predictions and the ground truth are aligned
> The Hungarian matching algorithm is widely used to align the predictions and the ground truth in the unsupervised setting. The objective is to find an assignment function that maps each predicted class to a ground truth class with the minimum total cost. One needs to first compute a cost matrix with each entry indicating the cost of an assignment and uses the Hungarian matching algorithm to find the best assignment function. In practice, we utilize the linear_sum_assignment function from the scipy package to calculate the assignment. This function takes the cost matrix as input and outputs a list of pairs, such as $[(m_0, n_0), (m_1, n_1), ...]$, where each predicted class $m_i$ is aligned with a ground truth class $n_i$.
>
> ### How were the other parameters chosen/fine-tuned
> We have conducted experiments to evaluate the sensitivity of the momentum parameter and the temperature parameter. Empirically, we found that using a large momentum parameter and a small temperature parameter consistently leads to good results on all datasets. Throughout the experiment, we set these two parameters as follows: momentum parameter $\tau = 0.1$ and temperature parameter $\alpha = 0.998$, unless stated otherwise.
>
> ### Explanation of “which represents discontinuities between image patches that should be preserved.”
> We apologize for any confusion caused. In STEGO, a correlation tensor $S$ (with each entry within $[-1, 1]$ when using cosine similarity) is used to measure the similarities between two image patches. A large value of $S_{ij}$ indicates that patches $i$ and $j$ have a high similarity and should belong to the same class. Conversely, a small value of $S_{ij}$, especially when $S_{ij} < 0$, indicates a low similarity between patches $i$ and $j$, suggesting they may be located at the boundary of segments and tend to belong to different classes. We refer to these differences between patches as "discontinuities". Under the smoothness assumption, an effective model should not only encourage piecewise smoothness but also maintain the discontinuities between image segments. Preserving these discontinuities is crucial to avoid trivial solutions. However, in STEGO, the negative part of the correlation tensor $S$ is discarded using a 0-clamp, potentially overlooking the significance of these discontinuities. In contrast, our label penalty function $\delta(·, ·)$, which satisfies $0 ≤ \delta(·, ·) ≤ 1$, possesses a desirable property compared to $S$. It allows us to properly handle and preserve these crucial discontinuities in the smoothness term, contributing to improved performance and meaningful segmentation maps in our approach.

---

> > ### Comment · Reviewer_gyDm · 2023-08-16
> >
> > Thank you, I'm happy with the response.
> >
> > This work leaves an overall positive impression and I intend to keep my score. I also read the other reviews. I agree with some points, but most of them seem to be addressed well and can be included in the camera-ready. In other cases, I tend to agree with the authors that they do not seem essential (e.g. linear probing) or even feasible (e.g. adding the smoothness loss to STEGO).

---

> > > ### Author Response · Authors · 2023-08-19
> > > **Thanks for your constructive feedback**
> > >
> > > We would like to thank the reviewer for the constructive feedback which helps shape our revision. Please let us know if there are any additional concerns or suggestions that could further enhance the quality of this work.

---

### Official Review · Reviewer_1NSo · 2023-07-10

**Soundness:** 4 excellent
**Presentation:** 4 excellent
**Contribution:** 2 fair
**Rating:** 5
**Confidence:** 2

**Summary:**

The paper introduces a new approach called SmooSeg for unsupervised semantic segmentation, which aims to segment images into semantic groups without manual annotation. SmooSeg is based on the idea of smoothness: adjacent features in a metric space should share the same semantics. Specifically, it formulates unsupervised semantic segmentation as an energy minimization problem (Eq. 1,4). Training is performed using teacher-student style self-training.

**Strengths:**

### The paper is well-structured.
* The introduction is well-written and motivates the problem well. The methods section breaks down the contribution into provides helpful preliminaries (S3.1) in a good amount of detail. It also describes the contribution (S3.2-3.4) precisely without unnecessary complications.

### The visualizations provided are nicely done
* The t-SNE plot in Figure 5 is quite illustrative, and the qualitative examples in Figure 3 are also helpful.

### Algorithm 1 does a good job of communicating the proposed method.
 * It is very helpful to be able to refer to the PyTorch pseudocode alongside the equations in the paper.

### The “Discussion with CRF and STEGO” section is very helpful
* My natural first question upon reading the introduction and the beginning of the methods section was about the relationship between SmooSeg and CRFs. This section answered some of my questions, as it gave an analysis of their relationship.

**Weaknesses:**

### It would help to have further analysis of why the method works.
* As discussed in the paper’s introduction, the idea of smoothness is image segmentation has a long history, having been explored since well before deep learning. This is especially true of energy minimization approaches, for which many different types of smoothness losses have been proposed over the years. Empirically, it seems that the performance of the smoothness loss proposed in this paper is good. However, I do not really understand _why_ it should be better than any other approach to enforcing image smoothness, such as the approach applied in CRFs. What would make this paper really useful to the community would be if it tried to understand why this particular formulation works well, so that the community can learn some generalizable lessons/insights (which could then be applied to other problem domains as well).
* Is it something particular about the combination of your smoothness term and the student-teacher approach that works well, or is it each of them individually?
* What would happen if you took STEGO (exactly as it is) and added your smoothness loss to their method? Would it be good, or does your smoothness loss work particularly well when combined with your student-teacher method?

### Why do you also need to apply a CRF?
* The supplement states: “We also use a CRF as the post-processing to refine the predicted semantic maps.” This confuses me. If you already have a smoothness loss (which is your contribution), why do you need a CRF?
* How does your performance compare without a CRF?

_Meta note on the score (because there is nowhere else to put this in the review)_: I am between 4 and 5. I gave a preliminary score of 4, but I am certainly willing to raise my score when the above questions/points are answered.

**Questions:**

### Removing E_{data} in Table 4
* For Table 4, I’m surprised that the method still works when the data term is removed in Table 4. In that experiment, are there any losses apart from the smoothness term? Without the data term, should there not be degenerate solutions?

**Limitations:**

Limitations are discussed adequately in the paper. One main limitation (how to set hyperparameters) is mentioned clearly at the end of the paper and addressed in the Appendix.

---

> ### Author Rebuttal · Authors · 2023-08-08
>
> Thank you for taking the time to review our work. Some of the concerns are responded in the "General Response" thread. Please kindly refer to the response regarding "**why this method works**" and "**CRF**".
>
> ### Roles of the smoothness term and the student-teacher predictor
> We apologize for any confusion caused by the lack of detailed information of our ablation study in the main submission. We will make it clear in our revision. The student-teacher predictor is specifically designed for the data term. The smoothness term requires a predictor but not necessarily a student-teacher one to generate the label map for label penalty.
> In order to justify our choice of the student-teacher predictor and the effectiveness of the smoothness term, we remove the student branch in the variant of “w/o $E_{data}$” (Table 4) and the data term in the loss function. That is, we only keep the prototypes in the teacher branch to generate the label map for the smoothness term. The prototypes $P^t$ and the projector are then updated simultaneously by SGD from the smoothness term. Our ablation study shows that the smoothness term with a normal predictor has some performance downgrade (see w/o $E_{data}$ in Table 4). On the other hand, the student-teacher predictor alone, without considering any smoothness prior, could not obtain reasonable results (see w/o $E_{smooth}$ in Table 4). Combining the smoothness term and the student-teacher predictor leads to the best performance.
>
> ### Add smoothness loss to STEGO
> The smoothness loss in our work is similar in form to the correlation loss in STEGO. However, the main difference is that the correlation loss aims to perform feature correspondence, while our smoothness loss works on the label penalty and requires a predictor to generate the label map. From this point, there is no straightforward way to add our smoothness loss to STEGO.
>
> ### (Question) Removing $E_{data}$ in Table 4
> In Table 4, we only used the smoothness term as the sole loss when we removed the data term. The smoothness term utilizes a closeness matrix constructed from high-level features of a pre-trained model, acting as strong supervision signals to guide label learning for all image patches. Therefore, it is reasonable to see that the smoothness term contributes significantly to the overall performance. On the contrary, the data term operates in a self-training fashion with pseudo labels derived from the teacher branch, which alone cannot generate accurate segmentation maps, as evidenced by the extremely poor results of "w/o $E_{smooth}$" in Table 4. When combined with $E_{smooth}$, which provides enhanced pseudo labels, the data term aiming to minimize the entropy of the predicted segmentation maps, yields performance improvements. We will update the descriptions in the ablation study to provide better clarity.

---

> > ### Comment · Reviewer_1NSo · 2023-08-18
> > **Response**
> >
> > Thank you for your response and clarifications.
> >
> > After reading your response and the reviews of the other authors, I am satisfied that the proposed method is sufficiently different from STEGO. I agree there is no straightforward way to add the smoothness loss to STEGO; apologies for the misunderstanding.
> >
> > I also now get that $E_{data}$ is essentially a self-training loss; it is good to see that the method works even without $E_{data}$.
> >
> > I will be updating my score from 4 to 5.

---

> > > ### Author Response · Authors · 2023-08-19
> > > **Thanks for your constructive feedback**
> > >
> > > We are glad that our response addressed all your concerns. We also greatly appreciate that you awarded us a higher score. Please let us know if there are any additional concerns or suggestions that could further enhance the quality of this work.

---

### Official Review · Reviewer_HQSk · 2023-07-10

**Soundness:** 3 good
**Presentation:** 3 good
**Contribution:** 3 good
**Rating:** 6
**Confidence:** 4

**Summary:**

The paper tackles the difficult task of unsupervised semantic segmentation at the level of the scene (dense prediction). The authors exploit the piecewise coherence regarding the semantics, texture, or color that similar objects naturally have (termed as smoothness prior). The problem formulation raises some challenges the author solves by using a pre-trained (frozen) feature extractor to model the closeness relationships among observations and introduce a novel pairwise smoothness loss, and a teacher-student style predictor. Results are showcased on popular benchmarks: COCOSttuff, Cityscapes, and Potsdam, on which the authors report state-of-the-art performance.

**Strengths:**

1. Originality - the problem formulation (energy minimization objective function) and the addition of the smoothness prior counts as novel work.
2. Quality - the paper is decently structured, method and experimental analysis are sound and convincing.
3. Clarity - some aspects of the paper could be improved. Some details are left out, making it be challenging to reproduce the results based on the information provided in the paper.
4. Significance - the topic is indeed relevant to the research community.

**Weaknesses:**

* L112 - The architecture description paragraph is tough to follow and the dedicated figure (Figure 1 in the main submission) is overcomplicated. Also, the use of the term "prototypes" for the teacher-student paradigm is confusing. Please consider rephrasing this part.
* L62 vs. L312 - minor contradiction.
* L307 - Type "We" - no capital letter
* The main submission has an appendix document but there is no reference in the main submission regarding the contents of the supplementary material.
* Important details are in the supplementary material and not mentioned in the current submission - such as the use of CRF for further refining the final segmentation maps.
* In the experiments section there is no mention of how the authors produced the segmentation maps from SSL feature extractors such as ResNet50, MoCoV2, DINO (Table 1), DINO (Table 2), or DINO, DINOV2 (Table 3).

**Questions:**

Please see weaknesses above.

**Limitations:**

The authors have not properly addressed the limitations of their proposed method. I highly encourage them to do so in a dedicated section in the main submission. Ideas for future work are a plus.

---

> ### Author Rebuttal · Authors · 2023-08-08
>
> Thank you for taking the time to review our work.
>
> ### Architecture description is tough to follow, the dedicated figure is overcomplicated，and the term "prototypes" is confusing.
>
> 1\) We apologize for the confusion on the architecture description and the figure (we guess you are referring to Fig.2 instead of Fig.1 here?). To improve clarity, we will itemize main modules/steps in the architecture description.
> - Step 1: feature extraction: $X = f_{\theta}(I)$, where $f_{\theta}$ denotes the extractor, a pre-trained model.
> - Step 2: feature projection: $Z = h_{\theta}(X)$, where $h_{\theta}$ denotes the projector, a non-linear projection head.
> - Step 3: label prediction: $A^{s,t} = Z^TP^{s,t}$ in both teacher and student branches in the predictor.
>
> We will also simplify Fig.2 by e.g., removing unnecessary annotations.
>
> 2\) We thank the reviewer for pointing out the confusion on "prototypes". We will clarify the term "prototypes" as "class centers" in the paper for better clarity.
>
> ### Minor contradiction and typos
>
> We greatly appreciate your careful reading and pointing out these errors. We will update the description in L312 to eliminate the inconsistency and correct any typos.
>
> ### No reference regarding the supplementary material
>
> Thank you for bringing this to our attention. We will make a reference for each Appendix in the corresponding section of the revised main paper as:
> - *L 215: “Implementation details can be found in Appendix A.”*
> - *L 249-251: “Additional qualitative results, along with color maps, can be found in Appendix C.”*
> - *L 304: “However, we present a feasible strategy in Appendix B to alleviate this issue.”*
>
>
> ### The use of CRF not mentioned in the submission
>
> Please see the general response.
>
> ### Implementation detail of baselines
>
> We will add the following details in the experiments section.
>
> *Results of ResNet50, MoCoV2, DINO (Table 1), DINO (Tables 1,2,3) are directly cited from the paper [8], while the results of DINOV2 (Table 3) are obtained by our implementation. For these baselines, we first extracted dense features for all images. We then utilized a minibatch k-means algorithm to perform patches grouping, which resulted in the final segmentation maps.*
>
> ### Limitations:
> Thanks for your suggestions. We will reorganize the materials and discuss more in the main manuscipt about the setting of hyper-parameters in the smoothness term. It is observed that the pre-calculation on the statistics for estimating the feature similarities could be a good prior in setting the hyper-parameters, which is still under investigation in our further study.

---

### Author Rebuttal · Authors · 2023-08-10

# General Response
We appreciate all reviewers for the highly constructive comments that help improve the paper quality. We also thank the reviewers for the recognition of our work being well structured (**1NSo**, **gyDm**) and well presented (**iJZU**, **yftt**), the proposed solution being simple (**gyDm**), novel (**HQSk**), experimentally convincing (**HQSk**, **R6c1n**, **RceoA**), insightful (**HQSk**), of sufficient scope (**gyDm**), and covering various datasets (**yftt**).

*We respond to some general comments as follows. We hope our rebuttal address the reviewers' questions and concerns. We would be more than happy to discuss with all reviewers if they still have any unresolved concerns or additional questions about the paper or our rebuttal.*

## Why our method works (@1NSo, iJZU)
**@1NSo** **The difference between our method and previous smoothness-based methods.** Thank you for the thoughtful feedback. As mentioned in the introduction, one of the key challenges in applying smoothness priors to unsupervised semantic segmentation is to define a good closeness relationship among image patches. The main difference of our proposed smoothness prior from those defined in other methods lies in the definition of “adjacent” patches/pixels. Early image segmentation methods, as well as CRF models, define adjacency in the coordinate space, e.g., using a 4-connected or 8-connected grid in the coordinate space to define the adjacent pixels of a given pixel. These methods primarily relied on low-level appearance information and fell short in capturing high-level semantic information in images. Consequently, these approaches alone may not yield good results for high-level semantic segmentation tasks, as it may struggle to effectively capture the complex relationship of images patches necessary for semantic understanding. Our smoothness prior, on the contrary, imposes the closeness in the high-level feature space generated by self-supervised representation learning methods. This definition better suits the task of semantic segmentation as it better captures high-level semantic similarity. Such a closeness design provides strong supervision signals to guide the label learning effectively. Through our ablation study, we demonstrate that the proposed smoothness prior alone (w/o $E_{data}$ in Table 4) can still achieve promising results. We remark that the design of “across images” smoothness term ($E_{smooth}^{across}$) further extends the definition of "smoothness" from local patches to cross-image levels and is also novel and effective (see the ablation in Table 4). We will add these discussions in the revision.


**@iJZU** **What the network is learning from.** (1) We apologize for not explicitly mentioning the use of `torch.no_grad` in Algorithm 1. We will include this missing detail in the revision. We did, however, state that we utilize a frozen pre-trained model as an extractor. (2) Our core learning mechanism revolves around employing the closeness matrices (by Eq (2)) as smoothness signals to facilitate the projector and predictor learning. These closeness matrices act as strong supervision cues to guide the label learning (projector + predictor). (3) In relation to Fig 5, your understanding aligns accurately. After learning the smooth labeling function, similar features are already assigned similar labels, and the smoothness penalty is reduced.

## CRF

- **Why need to apply a CRF (@1NSo):** We would like to emphasize that CRF postprocessing is a common practice in both supervised and unsupervised semantic segmentation (line 23, p6 in [8]), and that the use of CRF does not overshadow the contribution of the smoothness term in our work. The smoothness prior in this work performs on the high-level feature maps and mainly contributes to semantic smoothness, while CRF operates on pixels to refine the fine details and remedy the resolution loss caused by the final upsample operation that exists in most semantic segmentation models (a normal upsample rate is 8x8). Therefore, the application of a CRF serves as a supplement to our smoothness prior to further refine low-level smoothness.

- **The ablation of CRF (@1NSo, @yftt)**. The following table demonstrates that SmooSeg still achieves state-of-the-art without CRF, as CRF only accounts for $1.3$ mIoU out of the $2.2$ mIoU performance gain over STEGO. Moreover, we also present qualitative visualizations with and without CRF in the appended PDF file. It is found that CRF is able to refine the quality of fine details on both STEGO and SmooSeg. However, SmooSeg is consistently more semantically coherent than STEGO either with or without CRF.

    |                 | COCOStuff27 | Cityscapes  |  Potsdam-3  |    Avg.     |
    | --------------- |:-----------:|:-----------:|:-----------:|:-----------:|
    |                 | Acc / mIoU  | Acc / mIoU  | Acc / mIoU  | Acc / mIoU  |
    | STEGO w/o CRF   | 46.5 / 22.4 | 63.5 / 16.8 | 74.1 / 58.9 | 61.4 / 32.7 |
    | STEGO           | 48.3 / 24.5 | 69.8 / 17.6 | 77.0 / 62.6 | 65.0 / 34.9 |
    | SmooSeg w/o CRF | 60.6 / 25.2 | 79.8 / 18.0 | 81.4 / 68.4 | 73.9 / 37.2 |
    | SmooSeg         | 63.2 / 26.7 | 82.8 / 18.4 | 82.7 / 70.3 | 76.2 / 38.5 |

- **The use of CRF not mentioned in the main submission (@HQSk):** As previous studies (e.g., STEGO) reported their performance with CRF by default, we compare the performance with CRF applied during test by default as well. Thanks for pointing it out, and we will move the postprocessing details from the appendix to the main manuscript in the revised version.
- **CRF weight (@yftt)**. We utilized *pydensecrf* for our CRF refinement, with all parameters set to default values to make a fair comparison with previous studies.

---

> ### Comment · Reviewer_iJZU · 2023-08-21
>
> > We apologize for not explicitly mentioning the use of torch.no_grad in Algorithm 1. We will include this missing detail in the revision.
>
> Maybe not needed! Just mentioned it as one thing I had looked for, as a hint as to what was being learned. Definitely okay if the authors can clarify this same point some other way.

---

### Decision · Program_Chairs · 2023-09-21

**Decision:**

Accept (poster)

**Comment:**

While the initial reviews were mixed, the rebuttal and discussion cleared up the majority of the remaining concerns and extensively discussed STEGO, which was one of the main remaining concerns. Overall, while the paper reuses some ideas from STEGO (dimensionality reduction + clustering) it adds a smoothness constraint which shows a substantial performance improvement. It is thus of interest to the community and should be accepted.